


# Modulation of radiative aerosols effects by atmospheric circulation over the Euro-Mediterranean region

Pierre Nabat[1], Samuel Somot[1], Christophe Cassou[2], Marc Mallet[1], Martine Michou[1], Dominique Bouniol[1], Bertrand Decharme[1], Thomas Drugé[1], Romain Roehrig[1], and David Saint-Martin[1]

[1]CNRM, Université de Toulouse, Météo-France, CNRS, Toulouse, France
[2]CECI, Université de Toulouse, CNRS, CERFACS, Toulouse, France

**Correspondence:** Pierre Nabat (pierre.nabat@meteo.fr)

**Abstract.** The present work aims at better understanding regional climate-aerosol interactions by studying the relationships between aerosols and synoptic atmospheric circulation over the Euro-Mediterranean region. Two 40-year simulations (1979-2018) have been carried out with the CNRM-ALADIN64 regional climate model, one using interactive aerosols and the other one without any aerosol. The simulation with aerosols has been evaluated in terms of different climate and aerosol parameters.

This evaluation shows a good agreement between the model and observations, significant improvements compared to the previous model version, and consequently the relevance of using this model for the study of climate-aerosol interactions over this region. A first attempt to explain the climate variability of aerosols is based on the use of the North-Atlantic Oscillation (NAO) index, which explains a significant part of the interannual variability, notably in winter for the export of dust aerosols over the Atlantic Ocean and the Eastern Mediterranean, and in summer for the positive anomalies of anthropogenic aerosols

over Western Europe. This index is however not sufficient to fully understand the variations of aerosols in this region, notably at daily scale. The use of "weather regimes", namely persisting meteorological patterns, stable at synoptic scale for a few days, provide a relevant description of atmospheric circulation, which drives the emission, transport and deposition of aerosols. The four weather regimes usually defined in this area in winter and in summer bring significant information to answer this question. The blocking and NAO+ regimes are largely favourable to strong aerosol effects on shortwave surface radiation and surface

temperature, either because of higher aerosol loads, or because of weaker cloud fraction, which reinforces the direct aerosol effect. Inversely the NAO- and Atlantic Ridge regimes are unfavourable to aerosol radiative effects, because of weaker aerosol concentrations and increased cloud cover. This study thus puts forward the strong dependence of aerosol loads on the synoptic circulation from interannual to daily scales, and as a consequence, the important modulation of the aerosol effects on shortwave surface radiation and surface temperature by atmospheric circulation. The role of cloud cover is essential in this modulation as

shown by the use of weather regimes.

## 1 Introduction

In the climate system atmospheric aerosols exert a strong influence on the radiative budget and clouds (Kaufman et al., 2002). They absorb and scatter radiation (direct effect), but they also interact with cloud microphysics as cloud condensation nuclei





(indirect effect). Their numerous and various sources as well as their relatively short life time give them a high spatio-temporal variability, notably over the Mediterranean. In this region, aerosols accumulate coming from the industrial and urban sources in Europe, North African towns, biomass burning in Eastern Europe, desert sources in the Sahara, and directly from the Mediterranean Sea itself (Lelieveld et al., 2002; Basart et al., 2009). Thus aerosols have strong impacts on the European and

Mediterranean regional climate insofar as they can for example modify the radiative budget (Spyrou et al., 2013; Nabat et al., 2012), surface temperature (Zanis et al., 2012; Nabat et al., 2015b), past climate trends (Zubler et al., 2011; Nabat et al., 2014; Boé et al., in rev.) and air-sea fluxes (Nabat et al., 2015b). All these effects are generally more important in spring and summer, when maximal aerosol loads are observed. Indeed the dry season favours a longer residence time for atmospheric aerosols. Besides, dust outbreaks characterized by large plumes of Saharan desert dust particles, are more frequent in this season. As

a consequence, emissions, transport and deposition are different factors that can explain the high spatio-temporal variability of aerosols in the Mediterranean. For example, the daily variability of surface shortwave radiation and temperature is better represented in climate models when using a prognostic representation of dust aerosols (Nabat et al., 2015a).

Nevertheless the latter is also partly due to variations in atmospheric conditions. For example, Moulin et al. (1998) have shown that the location of the main low and high pressure systems drive the dust outbreaks occurring over the Mediterranean

basin. In spring, sharav cyclones over Algeria and Libya (Alpert and Ziv, 1989) are responsible for strong dust storms and an induced southwesterly flow bringing dust loads over the Eastern Mediterranean. Then from June, the settling of a high pressure system over Libya keeps the lows over Western Sahara, moving the flow to southerly and the dust loads to Central Mediterranean. During summer, these dust outbreaks are moving to the western basin, because of cyclogenesis phenomena over the Iberic peninsula. Regarding sea-salt particles, high loads are notably triggered by strong local winds such as mistral

and tramontane (Despiau et al., 1996), themselves favoured during northwesterly flows over Western Europe. In this study, the objective is to establish more systematically the relationships between atmospheric conditions and aerosol loads over the Euro-Mediterranean area from yearly to daily time scales.

Previous studies have used climate indexes such as the North-Atlantic Oscillation (NAO) to characterize aerosol variability over the Mediterranean. Moulin et al. (1997) and Papadimas et al. (2008) have found a positive correlation between the NAO

index and Mediterranean Aerosol Optical Depth (AOD) retrieved by satellite instruments, respectively MeteoSat and MODIS. A positive phase of the NAO generally leads to drier conditions over the Mediterranean, thus favouring high AOD over this region. The negative phase has the opposite effects. Between 2000 and 2006, Papadimas et al. (2008) have therefore related the decrease in Mediterranean AOD to a decrease in the NAO index during the same period. However, using model output, Ginoux et al. (2004) have not confirmed this positive correlation between NAO and Mediterranean AOD, but only the positive

correlation between NAO and dust export over the Atlantic Ocean.

"Weather regimes" (Vautard, 1990) provide another approach to study the link between aerosols and synoptic atmospheric circulation. They consist in persistent meteorological structures of pressure, wind and temperature, that embed synoptic scale event for a few days. They are generally defined from a statistical method of automated classification, generally based on the sea level pressure or the 500-hPa geopotential daily anomalies. The weather regime paradigm has the advantage for enhancing the

level of predictability of the atmosphere (Cassou, 2008). Weather regimes have been used to explain atmospheric variability at





synoptic scale in several processes such as European heat waves (Cassou et al., 2005), extreme precipitation (Sanchez-Gomez et al., 2008), cold extremes in Europe (Cattiaux et al., 2010) and deep water formation in the North-Western Mediterranean Sea (Somot et al., 2018). As aerosols are influenced by atmospheric conditions both for emissions (dust and sea-salt), transport and deposition, we could expect a strong modulation of aerosols radiative effects by weather regimes. The previous study of

Ménégoz et al. (2010) has focused on the interactions between weather regimes and aerosols over the North Atlantic European region, showing that dynamical processes impact the different aerosol burdens. However this study has used a model in which only three aerosol species were represented (sulphate, black carbon and desert dust) with climatological emissions for dust aerosols. Dust emission is indeed favoured by strong winds over Sahara, and dust transport over the Mediterranean Sea seems to be more frequent in southerly to southwesterly flows over this region, as shown in case studies (Nabat et al., 2015a). Thus

the regime of intense dust episodes in the Mediterranean area has been studied by Gkikas et al. (2013), who have highlighted this strong relationship between synoptic circulation at daily scale and these dust episodes.

As a matter of fact, the interactions between aerosol loads and atmospheric circulation at synoptic scale are very complex insofar as aerosol loads are strongly affected by meteorological conditions, and in the meanwhile these meteorological conditions are modified by the radiative and climate impacts of aerosols. Most of climate studies already published only consider

these interactions at yearly or seasonal time scales, while the daily time scale would be needed to better understand these interactions. That is the reason why the present work aims at establishing more completely these interactions between atmospheric circulation and aerosols, not only at yearly and monthly time scale using the North Atlantic Oscillation, but also at daily time scale using weather regimes. The approach used here relies on the use of a regional climate system model, which enables us to have an explicit representation of the main aerosol types (sulphate, organic matter, black carbon, dust, sea-salt, nitrates and

ammonium), their different processes (emission, transport, deposition) and their interactions with radiation and clouds in a climate regional model.

## 2   Methodology

### 2.1   The regional climate model: CNRM-ALADIN64

The present study is realized using the regional climate model ALADIN-Climat, called CNRM-ALADIN64 in the version used

in the present study. This model is based on a bi-spectral semi-implicit semi-Lagrangian advection scheme, and is used here in its version 6.4 with a horizontal resolution of 50 km and 91 vertical levels as in Drugé et al. (2019). This version is based on the cycle 37T1 of ARPEGE-IFS, and is very close to the version described in Daniel et al. (2018). All the parameterizations used in CNRM-ALADIN64 are summarized in Table1, separated between the atmosphere, the surface and the aerosol scheme. Compared to the previous ALADIN-Climat generation used in several studies such as Nabat et al. (2014) and Nabat

et al. (2015b), most of the atmospheric physics has been revisited (Voldoire et al., 2019). It now includes a convection scheme representing in a unified way dry, shallow and deep convection (PCMT, Piriou et al., 2007; Guérémy, 2011), a moist turbulence scheme based on a prognostic equation of the turbulent kinetic energy (Cuxart et al., 2000) and the large-scale microphysics scheme of Lopez (2002) which describes liquid and ice particles as well as rain and snow using prognostic variables. The





shortwave radiation scheme has been updated to six bands (Fouquart and Bonnel, 1980; Morcrette et al., 2008), while the longwave radiation scheme is based on the Rapid Radiative Transfer Model (RRTM, Mlawer et al., 1997).

With regards to the surface, CNRM-ALADIN64 uses the version 8 of the SURFEX modelling platform (https://www.umr-cnrm.fr/surfex/), including notably a tile approach which separates natural land surface, lake and sea areas in the calculation of surface fluxes. The sea-surface turbulent fluxes are derived from the ECUME (Exchange Coefficients from Unified Multi-campaigns Estimates) iterative approach (Belamari and Pirani, 2007). The lakes are represented using the bulk FLake model (http://www.flake.igb-berlin.de/) which computes the temporal evolution of the vertical lake temperature profile from the surface mixing layer to the bottom. More details of its use in SURFEX can be found in Le Moigne et al. (2016). The land surface is simulated using the ISBA-CTRIP coupled land surface modelling system (http://www.umr-cnrm.fr/spip.php?article1092lang=en) described in details in Decharme et al. (2019). To summarize, the ISBA (Interaction Soil Biosphere Atmosphere) land surface model computes the energy and water budgets at the surface-atmosphere interface while the CTRIP (CNRM version of the Total Runoff Integrating Pathways) river routing model simulates river discharge to the sea using the total runoff calculated by ISBA. An explicit two-way coupling between ISBA and CTRIP is used to represent (1) river flooding that interacts with the soil and the atmosphere through free-water evaporation, infiltration and precipitation interception and (2) water table into unconfined aquifers and upward capillarity fluxes into the superficial soil. Besides, CNRM-ALADIN64 is now using the XIOS Input/Output parallel server software (Meurdesoif, 2018), which facilitates the model workflow, especially on-line postprocessing and the production of output files in full netcdf format with appropriate attributes, in line with the Climate Model Output Rewriter (CMOR) format (Taylor and Doutriaux, 2004).

Finally, note that this version called CNRM-ALADIN64 is also used in the CORDEX framework, notably to contribute to the second phase of Euro-CORDEX and Med-CORDEX programs, as well as to the associated Flagship Pilot Studies on aerosols and air-sea interactions.

## 2.2 The aerosol scheme: TACTIC

CNRM-ALADIN64 includes a prognostic aerosol scheme called TACTIC (Tropospheric Aerosols for Climate In CNRM), which originally comes from the GEMS project (Morcrette et al., 2009), and which has been first used in the CNRM climate models in the studies of Michou et al. (2015) and Nabat et al. (2015a), and more recently in Watson et al. (2018) and Michou et al. (2019). In addition to the five main aerosol species (dust, sea-salt, sulphate, black carbon and organic matter) initially included, nitrate and ammonium particles have been recently added (Drugé et al., 2019). These aerosols are distributed in 16 prognostic variables, submitted to atmospheric processes (emission, transport and deposition). Sea-salt and dust emissions are dynamically calculated on-line as a function of surface wind and soil characteristics, while anthropogenic and biomass burning emissions are based on monthly inventories (see the following paragraph on simulations). Compared to the version used in Nabat et al. (2015a), the main changes, described below, are the implementation of a new sea-salt emission parameterization, a revision of the dust emission parameterization, and a correction in the treatment of aerosols in the lateral boundary conditions. More details about the other parameters which have not been modified can be found in Michou et al. (2015) and Nabat et al.





(2015a), as well as in Drugé et al. (2019) as far as nitrates and ammonium are concerned.

Sea-salt particles are represented with three size bins, whose original limits defined by Morcrette et al. (2009) were 0.03, 0.5, 5 and 20 $\mu$m. However, these limits do not correspond to the different processes at stake in the formation of sea-salt particles.
Indeed the smallest particles whose diameter is generally lower than 1 $\mu$m are film droplets produced from bubble bursting at the surface. The jet following the void left at the sea surface by the bubble leads to the formation of larger particles, typically between 1 and 10 $\mu$m. Besides, even larger particles can be produced in the presence of strong winds when spume is torn off the wave crests. These different sizes of sea-salt particles have already been documented in several studies (O'Dowd et al., 1997; Sayer et al., 2012). Therefore the size distribution of sea-salt has been adapted to these processes, setting the bin limits
to 0.01, 1.0, 10.0 and 100.0 $\mu$m. The respective optical properties (extinction coefficients, single scattering albedo and asymmetry parameter) have been recalculated following the Mie theory, taking into account the dependence on relative humidity. Following the recommendations of the review paper of Grythe et al. (2014), the so-called G13T parameterization given in this study has been included in TACTIC for the sea-salt emissions of the first two size bins. It has been shown to be the closest to observations compared to other sea-salt emission parameterizations. It also takes into account the observed dependence of
sea-salt emissions to the sea surface temperature (Jaeglé et al., 2011). However it cannot be applied to larger particles, for which the formulation of Andreas (1998) has been chosen.

With regards to dust emission, the parameterization used in TACTIC is based on the studies of Marticorena and Bergametti (1995) for the calculation of the saltation flux, Gillette (1979) for the sandblasting flux, Fécan et al. (1999) for the influence of
soil moisture, and Kok (2011a) for the distribution of the vertical dust emitted flux. These parameterizations are the same as the ones used in Nabat et al. (2015a), except from a few corrections in the calculation of soil textures from the silt, clay and sans contents, and the use of the soil characteristics (roughness length, fraction of bare soil, soil moisture) coming from the SURFEX module (Decharme et al., 2019), instead of the climatological values used in the previous version.

Finally note that the configuration of CNRM-ALADIN64 including the TACTIC scheme described above can be completed by the coupling of the Mediterranean regional sea. In this case, this fully-coupled regional climate model is called CNRM-RCSM6 (6th generation of the CNRM Regional Climate System Model for the Mediterranean study), already used in Darmaraki et al. (2019). CNRM-RCSM6 represents at high-resolution all the components of the regional water and energy cycle and their interactions.

**2.3  Regional climate simulations**

Two simulations using CNRM-ALADIN64 have been carried out over the 1979-2018 period, driven by the ERA-Interim reanalysis (Dee et al., 2011) providing atmospheric lateral boundary forcing at 6 hour frequency and sea surface temperature at daily frequency. The first one called ALD-AER thereafter includes interactive aerosols generated by the TACTIC scheme described above, and their coupling with radiation and clouds. The second one, called ALD-NO thereafter, does not have any





aerosols. The comparison between the two simulations is used to estimate the effects of aerosols on regional climate as a function of weather regimes. In both simulations, the historical evolution of greenhouse gases (GHG) is included following the yearly global averages of Meinshausen et al. (2017) for CO2, N2O, CH4, CFC12 and a CFC11-equivalent species that includes the effects of all the other GHG of the original data set (39 species). As in the previous version of the model, the total

5 solar irradiance forcing is also taken into account with yearly averages following Matthes et al. (2017). Ozone radiative forcing is included using ozone concentrations coming from historical simulations carried out with the global Earth System Model CNRM-ESM2-1 (Séférian et al., 2019). No land use land cover change is taken into account in CNRM-ALADIN64.

A spectral nudging method (Radu et al., 2008) has been included in both simulations in order to have observed large scales, thus keeping the true natural climate variability. The nudging is applied between the top of the model and 700 hPa (with a

10 relaxation zone between 700 and 850 hPa), to wind (vorticity and divergence), humidity, temperature and surface pressure. Note that the intensity of the nudging varies according to time frequency which is used for each variable: 6 hours for vorticity, 24 hours for temperature, humidity and surface pressure, and 48 hours for divergence. These parameters are the same as in the nudging applied in the simulation of Nabat et al. (2015a).

The domain of simulation, shown in Fig. 1, is close to the one used in Nabat et al. (2015a). It is an enlarged Med-CORDEX

domain in order to include the main aerosol sources affecting the Mediterranean region. Anthropogenic and biomass burning emissions are based here on the CMIP6 historical inventories, provided respectively by Hoesly et al. (2018) and van Marle et al. (2017). Since the ERA-Interim forcing used in this study does not have interactive aerosols and the domain is thus large enough to include all the main sources of aerosols affecting the Mediterranean region, no aerosol is included in the lateral boundary conditions used in this study.

## 20 2.4 Observations

In order to evaluate the model performance, different datasets of observations have been used in the present work and are briefly described in the following paragraphs.

### 2.4.1 Satellite data

Satellite data are essential to evaluate climate simulations given the spatial and temporal scales that they can cover. Regarding

aerosols two instruments are widely used to evaluate aerosol optical depth: the MODerate resolution Imaging Spectroradiometer (MODIS) and the Multiangle Imaging SpectroRadiometer (MISR). The first one is a multi-spectral radiometer providing retrievals of aerosol microphysical and optical properties. From the recently released collection 6.1 from the Terra and Aqua platforms (Sayer et al., 2014), the merged datasets between the standard and deep target algorithms are used in the present study over the 2000-2017 period. This product covers the whole domain of simulation at 1°resolution, including the Sahara

desert. MISR, onboard the Terra platform, is also a passive radiometer providing data at four different wavelengths and nine directions, both over land and ocean. Level-3 monthly aerosol optical depths (MIL3MAE) at 0.5°resolution are used in this study over the 2000-2017 period.

Observed cloud fraction is documented using observations from the Cloud Aerosol Lidar with Orthogonal Polarization (CALIOP,



Winker et al., 2007) lidar onboard CALIPSO and Cloud Profiling Radar (CPR, Im et al., 2005) onboard CloudSat both flying in tandem in the A-Train constellation combined in the Radar Lidar geometrical profile product (Mace and Zhang, Journal of Geophysical Research). This combination is used in order to benefit from the ability of the CALIOP lidar to detect thin clouds and the CPR in penetrating deeper into opaque clouds as may be encountered in the Euro-Mediterranean area. Because of its

increased horizontal resolution several lidar samples are present in a radar profile and the radar lidar product provides then a lidar cloud fraction in each radar bin. For model comparison purpose, a cloud fraction is computed from this observational data set in each ALADIN model grid point as the fraction of the grid covered by a cloud detected in radar geometrical profile where the cloud mask is higher than 20 (corresponding to less than 16% of false detection) or when the lidar cloud fraction exceed 50% in a given bin. These thresholds were initially proposed and validated by Mace et al. (2009) and no attempt has been

made to identify precipitation. The cloud cover is then computed on an instantaneous basis for three atmospheric layers located below 3.2 km height, between 3.2 and 6.5 km height and above 6.5 km. Because of the low repetitivity of the A-Train, the data are accumulated at monthly time scale. Due to the CPR failure in April 2011 and its partial sampling (day time only) after 2012, the data used in this study are limited to the 2006-2011 period (4,5 years) and are originating of the CloudSat GEOPROF products Release 05.

At the top of the atmosphere, SW and LW radiation is evaluated against the Clouds and the Earths Radiant Energy System (CERES) - Energy Balanced and Filled (EBAF) data (Loeb et al., 2009), in the version 2.8 at 1°resolution over the 2000-2016 period.

### 2.4.2   Ground-based measurements

Ground-based observations are also used to evaluate the CNRM-ALADIN64 simulations. The AERONET network (Holben

et al., 1998) provides measurements of aerosol optical depth within automatic sun/sky radiometers. For this study daily average quality-assured data (Level 2.0) have been used from 72 stations distributed over the whole domain of the simulation (shown in Fig. 1). These stations have been selected because of their temporal period (at least 4 years of data) and their location (in order to have a spatially equally-distributed selection).

The Baseline Surface Radiation Network (BSRN) provides surface radiation measurements as well as other parameters such

as temperature and humidity over different stations around the world. These measurements from 10 stations over Europe and Africa have been used in this study (shown in Fig. 1). They are known for the quality of their data over long periods of time (Ohmura et al., 1998).

Land near-surface temperature and precipitation are compared to the version 4.0 of the Climatic Research Unit (CRU) dataset (Harris et al., 2014), provided by the University of East Anglia. This gridded product, whose horizontal resolution is 0.5°, is

based on land weather stations around the world.



## 2.5  Classification in weather regimes

Mid-latitude atmospheric circulation can be characterized through the positions of low and high pressure quasi-stationary large-scale systems, that drive higher frequency synoptic perturbations and associated winds and rainfall over the North Atlantic region. These structures are relatively stable during several days, and influence weather conditions in Europe and the Mediterranean region beyond meteorological time-scale. Such persisting conditions in pressure, leading to specific continental-scale wind, temperature and precipitation anomalies are called "weather regimes". They can be statistically defined using automated classification methods (Vautard, 1990; Cassou et al., 2004).

In the present work this paradigm has been applied to the ALADIN simulations, based on the daily anomalies of sea level pressure separately for winter (DJF) and for summer (JJA). These anomalies are classified using a k-means partition algorithm (Michelangeli et al., 1995) in order to get the decomposition of large-scale atmospheric conditions, respectively for winter and summer. Four weather regimes are consistently retained in our study following earlier literature (e.g. Cassou et al., 2004).

## 3  Evaluation of the ALD-AER simulation

Before studying the aerosols and their impact on climate variability, an evaluation of the main simulation ALD-AER is performed in this section in order to ensure the robustness of the subsequent results. Note that the ALD-NO simulation, which is similar to ALD-AER except for the aerosols, is not evaluated here, since such a couple of simulations had already been the focus of Nabat et al. (2015b).

### 3.1  Mean climate

First of all, an evaluation of atmospheric dynamics is presented in Figure 2. The comparison with sea level pressure (Fig. 2a) reveals that ALD-AER is close to ERA-Interim (bias less than 1hPa in most of the domain both in winter and summer), showing its capacity to reproduce the general circulation pattern over this region. This is eased by the spectral nudging method, but the latter is not applied below 850 hPa. Nevertheless, surface wind is slightly overestimated over the north of the Atlantic Ocean, especially in winter when the bias can reach 5 km/h on average.

Figure 3 presents the biases of the ALD-AER simulation in terms of land near-surface temperature, precipitation and cloud cover (total and low fractions) for winter (DJF) and summer (JJA). In parallel, the averages of these biases have been calculated on the six subregions of Europe and the Mediterranean (see Figure 2), defined in the frame of the PRUDENCE project Christensen and Christensen (2007) and already used in several studies (e.g. Kotlarski et al., 2014). These averages are presented in Table 3, together with the equivalent range found in Kotlarski et al. (2014) for the Euro-CORDEX ensemble of regional simulations over Europe. This evaluation of the new version 6.3 of the ALADIN-Climat model is also to be compared with a similar work carried out in Sevault et al. (2014) and Nabat et al. (2015b) with a previous version of the model. Note that the version 6 of ALADIN-Climat has been already evaluated over France in Daniel et al. (2018).

In winter (DJF), a residual cold bias is noticed in Europe (on average -0.5°), but significantly reduced compared to the ver-



sion 5 except in the higher grounds, notably in the Alps, and also to the Euro-CORDEX models analyzed in kotl14 who have strong biases for most of them (see Table 3. This cold bias is a little more pronounced over Northern Africa (-0.9°). Winter precipitation are overestimated in most part of Europe, +0.6 mm/day on average which represents a bias of 34%. However, this bias is in line with the bias of other regional climate models from Euro-CORDEX used in Kotlarski et al. (2014). As shown in

Table 3, the bias in winter precipitation is included in the range of biases of the other models in all 6 subregions. Besides, cloud cover is significantly improved in Europe, as the bias is only -1% on average. As in Nabat et al. (2015b), Northeastern Europe is affected by an overestimation of cloud cover, which is mainly due to low-level clouds as the bias is similar in low-level cloud fraction (Figure 3d). On the contrary, this parameter is underestimated over the Mediterranean Sea (-9%).

In summer (JJA), Europe is affected by a generalized warm bias (1.3°on average), which only affected Eastern Europe in
the previous version. Compared to Euro-CORDEX models, this bias is in the upper range as several models had on the contrary a cold bias. Over Northern Africa, a similar warm bias is noted (0.9°on average). This warm bias is consistent with an underestimation of cloud cover, especially in Western Africa (along the monsoon domain), and in Eastern Europe. Summer precipitation is also underestimated on average in Europe (-0.6 mm/day, that is to say -32%). In terms of radiation, Figure 4 shows the average biases of SW and LW fluxes at the top of the atmosphere against the CERES data. While in the previous
version of ALADIN (Nabat et al., 2015b) SW TOA radiation was affected by a large negative bias, it is better represented in winter in the present version (less than 1 W m$^{-2}$ for the average DJF bias in Europe and 3 W m$^{-2}$ for the Mediterranean Sea). In summer the bias over Europe is also reduced (only 3 W m$^{-2}$ on average) but the Atlantic Ocean is concerned by a large negative bias. With regards to LW radiation, as in the previous version of ALADIN, the bias at the TOA is small and uniform over the domain in winter (-5 W m$^{-2}$ on average in Europe) and in summer (-8 W m$^{-2}$ on average in Europe).

At the surface (Figure 5), radiation is evaluated using 10 BSRN stations throughout the domain of the simulation. For most of them, surface SW radiation (SSR) is overestimated by ALD-AER especially in summer when the bias ranges from 5 to 20 Wm$^{-2}$. This overestimation is likely due to the underestimation of cloud cover mentioned previously, and had already been noted in the previous version of the model evaluated over the Mediterranean in Sanchez-Gomez et al. (2011). However, for the few stations located in the south of the domain (Tamanrasset in Algeria, Sede Boker in Israel), the average bias is weaker.
Surface downwelling LW radiation is slightly underestimated, with a bias lower than 10 Wm$^{-2}$ on average.

In conclusion to this section, it has been proven that this new version of ALADIN-Climat is able to represent the main properties of mean regional climate over the Euro-Mediterranean area, in relative good agreement with observations, and with significant improvements compared to its previous version. The residual biases are not stronger than those of other regional models over Europe (Kotlarski et al., 2014). Note that this evaluation is not the main scope of this paper, and that more detailed
evaluation has already been published for the previous version of the model with regards to decadal variations (Dell'Aquila et al., 2018), daily precipitation (Fantini et al., 2018), and hydrometeorological extremes (Panthou et al., 2018).



## 3.2 Aerosols

### 3.2.1 General evaluation

The aerosols simulated by the TACTIC scheme in CNRM-ALADIN64 are evaluated against satellite and ground-based measurements. Figure 6 shows the average total AOD simulated in ALD-AER, as well as the AOD for each aerosol type. The
main spatial patterns of each aerosol type are consistent with their respective sources, namely a strong maximum in dust AOD over the Sahara, high sea-salt AOD over the Atlantic Ocean and to a lesser extent over the Mediterranean and Black Seas, and locally high values of sulphate, nitrate and organic matter AOD in Europe. Satellite data (MODIS and MISR), which provide only total AOD, have similar AOD spatial distribution. The spatial correlation between ALADIN and MODIS is 0.75, and 0.84 with MISR. However, discrepancies have been found locally, for example in the Benelux and in the Po Valley where
ALADIN AOD is overestimated compared to MODIS and especially MISR. This bias is much smaller than the negative bias in the previous version of the model which did not include nitrate aerosols (Drugé et al., 2019). Besides, sea-salt aerosols are also probably overestimated over the Northern Atlantic Ocean compared to MODIS and MISR, consistently with the surface wind overestimation described in the previous paragraph. Over the Mediterranean where dust particles are prevailing, total AOD simulated by ALADIN (0.18) is closest to MODIS (0.20) and MISR (0.16). Similar performance is noted over Northern
Africa (0.27 for ALD-AER, 0.33 for MODIS and 0.34 for MISR).

### 3.2.2 Confronting the model to station aerosol measurements

In order to further elaborate on the aerosols simulated by CNRM-ALADIN64, nine subregions have been defined on the domain (see Figure 1, to separate the influence of different aerosol sources. Each of them contains eight AERONET stations. The
AOD annual cycle simulated in ALD-AER is evaluated against one representative station of each subregion (Figure 7), and the AOD daily distribution is shown in Figure 8 for all stations. For each AERONET station, only the days where observations are available have been taken into account in the model both in Figures 7 and 8.

Over the western part of the domain (region A), sea-salt aerosols generated over the Atlantic Ocean are prevailing with a maximum in winter, as shown by the AOD distribution in the Azores station (Figure 7). This maximum is overestimated in winter,
especially in December in the Azores, probably in relationship with the overestimation of surface wind in the Atlantic Ocean. Both ALADIN and AERONET data show a decrease in AOD between winter and summer but the decrease in AOD is too strong in the model, yielding an underestimation of AOD in summer. This could be due to an underestimated transport of dust aerosols from the Sahara in this season. With regards to the AOD daily distribution in this region, the 90% interval (between the 5th and 95th percentiles) in ALD-AER is close that found in the observations for most stations (Figure 8). In addition,
ALD-AER also represents the average AOD values higher than the median as in the observations, due to the contribution of days with very high AOD, even if the model overestimates the most extreme AOD values in several stations.

Over Northern Africa and the Middle-East (regions B and C), dust aerosols have the most important contribution to AOD as seen in Tamanrasset and Solar_Village (Fig. 7), generating AOD higher than 0.40 in spring and summer. ALD-AER correctly





captures the annual mean and the seasonal cycle, despite a small underestimation at the end of the spring. In Solar_Village, sulphate aerosols also have a low contribution to AOD (around 0.05). The AOD daily distribution is also in general good agreement with observations in this area (Fig. 8), as well as the extreme AOD values. An exception is however noticed for stations 9 (Capo Verde) and 10 (Dakar) under the influence of dust aerosol exports in the Saharan Air Layer in which AOD is

5 underestimated, and for stations 17 (El Farafra) and 18 (Cairo) in Egypt where AOD is slightly overestimated.

In Southern Europe (regions D, E and F), AOD is mainly dominated by nitrate, sulphate and dust particles, with a small contribution of sea-salt aerosols in winter (Figure 7). The annual cycle is less pronounced than in the previous regions, but a maximum in spring and summer is also noted both in the model and in the observations, associated respectively with nitrate and dust aerosols. The AOD daily distribution is also well captured by ALD-AER, both for the median and the 90 % interval,

especially in regions D and E. The only exception is Montsec (station 30), where ALADIN overestimates the aerosol concentrations. This might be due to a mismatch between the model orography and the true altitude of the station located at 1574m. In some stations in region F in Greece and Turkey (44, 46, 47 and 48), ALADIN has a weak negative AOD bias, probably also due to a lack of transported dust in summer, or to underestimated local anthropogenic sources.

In regions G (Southern France, Northern Italy, and the Alps) and H (continental and Eastern Europe), anthropogenic aerosols

are prevailing as seen for example in Carpentras and Kyiv (Figure 7). The annual cycle is limited to an increase of nitrate and ammonium particles in spring, and to organic matter in summer in Eastern Europe due to biomass burning emissions. The model is also able to simulate various AOD daily distribution, such as the small range in Modena (station 50) or the larger range in Ispra (station 54). The cleaner air observed in the Alps in Davos (station 53) is also simulated in ALD-AER.

Finally, Northern Europe (region I) is also dominated by anthropogenic particles, with a higher contribution of nitrate aerosols,

probably overestimated in ALD-AER as already seen in Central Europe. The annual cycle is characterized by a maximum in spring as seen in Cabauw (station 66, Figure 7). The AOD daily distribution is in good agreement in AERONET and ALD-AER in several stations, for example Dunkerque (station 65) and Gotland (station 71), but the overestimation of nitrates disrupts this distribution in other stations such as Cabauw (66), Hamburg (68) and Leipzig (70). This overestimated contribution of nitrates had already been documented in Drugé et al. (2019).

To summarize, ALD-AER reasonably captures the annual cycle and daily distribution of AOD. Some discrepancies have also been emphasized, notably the overestimation of nitrates in spring in Northern Europe and the underestimation of the dust transport in summer in the Atlantic and in Southeastern Europe. However, these small biases do not prevent the model from being able to be used in this study to understand the variations of aerosols at the daily scale, and their potential impact on regional climate.

**4    Relations between aerosols and the North-Atlantic Oscillation**

In order to improve our understanding of the aerosol effects on the Mediterranean climate and their relationships with the atmospheric circulations, the present section analyzes how the North Atlantic Oscillation can modulate aerosol concentrations, first in winter and subsequently in summer.



## 4.1 In winter (DJF)

The respective correlation by season at yearly scale between the NAO index provided by the National Oceanic and Atmospheric Administration (NOAA), and the AOD (total, sea-salt, dust and sulphate) in ALD-AER, is presented in Figure 9 for winter and summer, while averages of these correlations over six regions of the domain are presented in Table 4. Note that the linear trend

of AOD has been removed from all the datasets. The NAO index is based on the surface sea-level pressure difference between the Subtropical (Azores) High and the Subpolar Low, and is thus assumed to be faithfully reproduced in ALD-AER, as it is laterally driven and spectrally nudged to the ERA-Interim reanalysis (Sanchez-Gomez et al., 2009). In winter, the correlation between total AOD and NAO has a zonal spatial distribution, characterized by positive values above 45°N (for example 0.44 in EURNW) and below 30°N (0.77 in AFRW), and negative values between these two limits (-0.15 in EURSE). Note that however

the area with significant values is larger for the positive correlations than for the negative ones. This pattern is consistent with the position of storm tracks and precipitation associated to winter NAO, namely an increase of storms and precipitation in Northern Europe during the positive phase, and a southward shift of the storm track in the negative phase (Pinto et al., 2009). Sea-salt aerosols logically follow this pattern, since emissions mostly depend on surface wind north of 30°N. The correlation between the NAO index and sea-salt emissions shown in Figure 10 confirms this pattern. Below 30°N, the positive correlation

is associated to dust particles, which explain a large part of the correlation between total AOD and the NAO index. It is the part of the domain where the correlation is the highest (between 0.7 and 0.9). This increase in dust AOD in the positive phase of NAO could be due to the reinforcement of eastern winds in Northern Africa generating more dust emissions (see Figure 10), due to the increase of surface pressure in the subtropics (Azores) and the associated geostrophic circulation.

With regards to sulphate aerosols (and more generally anthropogenic aerosols which are not shown here), a large positive

correlation is noted over the South of the domain, probably due to a higher residence time favoured by high pressures and reduced precipitation in positive NAO conditions. However this pattern is not noted in total AOD due to the weak sulphate concentrations in winter in this area.

Figure 11, which presents the interannual time series of the NAO index (in winter and summer) compared to the averaged AOD anomalies over three of the six regions defined in Figure 9 (EURNW, EURSE and AFRW), allows a better understanding

of these correlations. Associated yearly temporal correlations between the NAO index and the AOD anomalies are given in Table 4 for winter and summer. In AFRW, off the Western African coast, most of the years with large positive AOD anomalies (up to 40% compared to the averaged AOD in DJF) are also years with positive anomalies in NAO, such as 1983, 1989, 1995, 2000 and 2016, and conversely for negative anomalies such as in 1996 and 2010. As mentioned previously, previous studies (e.g. Moulin et al., 1997) had already highlighted such a correlation, since a positive phase of the NAO should favour

the export of dust aerosols over the Atlantic Ocean, due to the circulation induced by the strengthening of the Azores High and associated trade winds. This result is confirmed on the shorter period 2001-2017 when satellite data are available, as the respective correlations of ALADIN, MODIS and MISR are 0.82, 0.75 and 0.76. On the contrary, a negative correlation is noted over the Eastern Mediterranean area both in ALADIN and in satellite data, as shown in the EURSE domain. For example, the period 1999-2008 associated with positive NAO index is concerned by negative AOD anomalies in EURSE, and vice-versa





for the period 2009-2011. However, this is not the case on the whole period 1980-2017, and even if the sign of the correlation is in agreement in the three datasets, it is not significant at the level 0.05 except for MISR (-0.65). Moreover, the anomalies shown in Figure 11 are generally weaker than in AFRW. This absence of statistic significance is consistent with the results of Ginoux et al. (2004) who have not shown a relation between Mediterranean AOD and NAO, but not with those of Papadimas

et al. (2008) which have probably used a too short period of MODIS data. The same conclusion can be drawn for the region ALPS, where the correlation is negative but not significant in two out of the three data sets. In the two regions located above 45°N (EURNW and EURN), the correlation is largely positive in ALADIN (respectively 0.44 and 0.31), but weaker in MODIS and MISR where the threshold of significance is not reached. However, in these regions where clouds are prevailing in winter, the lack of AOD data retrieved by MODIS and MISR could hamper the estimation of this correlation with NAO. Finally in

EURSW, no significant correlation neither in ALADIN nor in satellite products has been found.

## 4.2   In summer (JJA)

In summer, the interannual variability of AOD is less important than in winter as noticed in Figure 11. The correlations between AOD and the NAO index are also weaker (Figure 9e), with fewer significant points at the level 0.05 than in winter. However, some regions such as a part of the Atlantic Ocean, Southern Europe and the Mediterranean Sea still have a significant corre-

lation between NAO and total AOD. In the Atlantic Ocean, the correlation is negative because of the contribution of sea-salt AOD, showing that the path of strong winds associated to the positive phase in summer is located at higher latitudes than in winter. This pattern is confirmed by the correlation between sea-salt emissions and NAO (Figure 10). In Southern Europe and the Mediterranean, the positive correlation is associated to the contribution of sulphate aerosols (and other anthropogenic particles to a lesser extent, not shown), since more stable conditions with large-scale subsidence and less rainfall is noted in

summer under the positive phase of NAO.

However when averaged over the six different regions studied, most of the correlations are not significant. When considering the interannual time series (Figure 11), it is indeed more difficult than in winter to conclude on the NAO impact on AOD. For example, the NAO index series is characterized by a long negative period between 2007 and 2016 (except 2013), which is not really the case in the time series of AOD in EURNW and EURSE (except the period 2008-2012 in EURSE). The averaged

correlations in Table 4 also show that they strongly depend on the period chosen: in ALD-AER the significant correlations of EURSE (0.33) and ALPS (0.34) over the 1979-2017 are much lower over the 2001-2017 period. In satellite data, no correlation is significant except for MISR in EURNW.

As a summary, the NAO index explains a significant part of the interannual variability of aerosols, notably in winter for

the export of dust aerosols over the Atlantic Ocean and the Eastern Mediterranean, and in summer for the positive anomalies of anthropogenic aerosols over Western Europe. Compared to existing literature, this study has further investigated these relationships between the NAO index and aerosol loads using a longer time period (1979-2018) and a detailed analysis by aerosol type. However, the significance of the correlations between AOD and NAO strongly depends on seasons and regions, and therefore this index is not sufficient to explain the whole variability in aerosol loads and their effects on regional climate





over the Mediterranean. The following section will therefore move to the daily time scale using the methodology of weather regimes, in order to further analyze these aerosol effects on regional climate.

## 5 Aerosol effects and weather regimes

As mentioned previously, weather regimes are an appropriate methodology to explain climate variability in Europe at daily
time scale. The objective of the present section is to understand if they can also explain the variability in aerosols and their impact on regional climate. The methodology described in Section 2 is applied to the ALADIN simulations in winter (DJF) and in summer (JJA), and results are presented below.

### 5.1  Definition of weather regimes

When applying the classification method to the daily sea level pressure output of the ALD-AER simulations, the usual four
main weather regimes are identified both for winter and summer : Atlantic Ridge (AR), negative NAO (NAO-), blocking (BL) and positive NAO (NAO+). The latter is close to the Atlantic Low (AL) regime identified for summer whose anomalies are less intense and northwestward shifted. The associated anomalies in sea level pressure are shown at the top of Figure 12 for winter and Figure 13 for summer.

The AR regime is characterized by a high pressure system over the Atlantic Ocean, rejecting low pressure systems over
Northern Europe. This pattern induces a northwestern flow over Western Europe and the Northwestern Mediterranean, which favours local winds such as mistral and tramontane. In the NAO- regime, the Icelandic low pressure system moves to the South, shifting the path of the Atlantic Jet and surface strong winds to southern latitudes. Thus the Mediterranean is affected by more cyclonic systems in this regime. On the contrary, in the winter NAO+ and summer AL regimes, the Mediterranean weather is drier and warmer, as the low pressure systems are rejected to the northern latitudes, while southern latitudes experience
positive anomalies of sea level pressure and large-scale subsidence. Finally, the BL regime is characterized by the presence of high pressures over Northwestern Europe.

Before analyzing each regime, note that weather regime anomalies are stronger in winter than in summer (see Figures 12 and 13), emphasizing that weather regimes are more significant in winter.

### 5.2  Aerosols and their effects as a function of the weather regimes

The anomalies of aerosol optical depth for each aerosol type and total are presented in Figures 12 and 13, while the anomalies of the aerosol impact on SSR (clear and all-sky) and temperature are shown in Figures 14 and 15. This impact of aerosols on SSR and surface temperature is calculated as the difference between ALD-AER and ALD-NO in SSR and surface temperature respectively. The anomalies of this impact are then calculated for each regime compared to the mean impact.





### 5.2.1 The blocking (BL) regime

As implied by its name, the BL regime is characterized by a synoptic situation where high pressure prevent low pressure systems from reaching Europe. This regime occurs both in winter and in summer, even if its intensity is weaker in summer. The first consequence of these high pressures over Northern Europe is to reduce the activity of storms over the Atlantic Ocean,

thus decreasing sea-salt emissions and thereby sea-salt AOD over Western and Northern Europe. On the contrary, over land, the atmosphere is more stable in this regime, thus allowing aerosols to live longer without being removed by wind, clouds, or precipitation. The sulphate AOD anomaly in the ALD-AER simulation is thus positive over most of Europe, in winter and in summer. The only exception concerns the eastern part of the domain in winter. A likely explanation for this region is that the drier air brought by high pressure tends to decrease relative humidity in the lower troposphere, thus decreasing the

aerosol extinction. With regards to dust aerosols, a small positive anomaly is observed over Northern Africa, probably due to the reinforcement of eastern winds in this region, induced by the circulation. In total, the AOD anomaly is positive over continental Europe and the Northern Mediterranean due to sulphate aerosols, and negative over the western part of the domain due to sea-salt particles. The pattern is quite similar in winter and in summer, with a larger extent of the positive anomaly in summer.

These AOD anomalies have an impact on SSR as seen in Figures 14 and 15. The clear-sky SSR pattern is very similar to that of AOD, with a decrease over continental Europe, and an increase over the near Atlantic Ocean. In all-sky SSR, only the negative anomaly over continental Europe remains, and even more widespread both in winter and in summer. Indeed as shown in Figure 12 cloud cover is reduced over Europe in the BL regime enabling more effects of aerosols on radiation, while over the Atlantic Ocean, the high values of cloud fraction limit their effects. As a consequence, a cooling effect of aerosols is noted over Europe

in the BL regime, reaching -0.2° in winter over the Po Valley and +0.2° in summer over Europe.

In order to better understand this impact of aerosols on climate in the different weather regimes, Figures 16, 17 and 18 show respectively the probability distribution functions (black lines) and the associated anomalies (colors) for each weather regime in winter, of the aerosol impact on SSR, the cloud cover, and the aerosol impact on surface temperature, in function of AOD for the different regions (one region per line) presented in Table 4. Figures 19, 20 and 21 are the summer equivalents. The

objective of these figures is to identify the change in the daily distribution of aerosol impacts in each regime compared to the average distribution. The latter, which is shown with black lines, is the same for each line (one line represents one region). For example in winter in EURNW, the distribution of SSR vs AOD (Figure 16) shows a maximum of occurrence of an impact of aerosols on SSR by -3 W m$^{-2}$ with an AOD of about 0.15. Concerning the anomalies of the BL regime in this region, more frequent days with relatively small AOD (between 0.10 and 0.25) are noted but with more impact on SSR (between -4 and -12

W m$^{-2}$), and thus inducing a stronger cooling (between -0.2 and -0.4°) than average in this region. This is made possible by the higher frequency of days with lower cloud cover (less than 80%). Similar behaviours can be identified in EURN. However, this process is reinforced in summer in EURNW, when during days with cloud cover lower than 50%, the decrease of SSR by aerosols is stronger by 10 W m$^{-2}$, and the induced cooling by 0.3° compared to the average aerosol effects. In the three other regions (EURSW, EURSE and ALPS), cloud cover is on average lower, so that the positive AOD anomalies in the BL





regime lead to stronger aerosol effects on SSR and surface temperature. For example, in winter EURSW, aerosols can generate a cooling of -0.4° in the BL regime against only -0.2° on average.

To summarize, in the BL regime, the aerosol effects are stronger over the whole Europe, either because they are more efficient due to the decrease of cloud cover (in Northern Europe), or because their concentrations is higher due to the more stable
conditions (Southern Europe).

### 5.2.2 The winter NAO+ and summer Atlantic Low (AL) regimes

The NAO+ regime in winter is characterized by a reinforcement of the Icelandic low pressure system, together with a positive pressure anomaly in Southern Europe. In summer, the equivalent regime (AL) has a similar negative pressure anomaly over the Atlantic, but further south, and a positive anomaly over Europe reaching higher latitudes than in winter. These conditions induce
a contrast in the AOD anomaly between Northwestern Europe and Southern Europe. On the one hand, sea-salt emissions are reinforced in the Northern Atlantic Ocean and Northern Sea in relation with the increase in surface wind, causing an increase in sea-salt and total AOD in this region (+0.05 on average in EURNW). Anthropogenic AOD over Northwestern Europe is however reduced, because of the excess precipitation under these conditions. On the other hand, sea-salt emissions are reduced further south in the Atlantic Ocean, because of high pressure inducing a decrease in surface wind. Over the Central and Eastern
Mediterranean, drier conditions allow a slight increase of sulphate AOD in winter. With regards to dust aerosols, in winter they contribute to a negative anomaly in total AOD in the Eastern Mediterranean, and to a positive anomaly off Northwestern Africa. All these patterns are consistent with the patterns described previously in the positive phase of NAO.

As in the BL regime, these AOD anomalies have impacts on SSR, especially in clear-sky conditions, for example in winter (Figure 14) with a negative anomaly over Northwestern Europe, Northern Europe and off Northwestern Africa (between -5 and
20   -20 W m$^{-2}$), and a smaller positive anomaly over the Eastern Mediterranean (between 2 and 5 W m$^{-2}$). However, in all-sky conditions, the negative winter anomalies in Northern Europe are not preserved, probably because of the important cloud cover (see Figure 12) that moderates the direct aerosol effect. As a consequence, the aerosol impact on surface temperature in this regime does not show any significant anomaly in winter. In summer (Figure 15), when cloud cover is lower, the differences between all-sky and clear-sky conditions are reduced, but the anomalies in the impact of aerosols on surface temperature remain
negligible, and spatially uncorrelated to the AOD anomalies.

However, the study of density probability functions of the aerosol impacts in Figures 16 to 21 show more interesting patterns. First, they confirm the fact that the increase in AOD in Northern Europe (EURNW and EURN) in winter does not have any impact on SSR (Figure 16) and temperature (Figure 18), because this increase in AOD occurs in very cloudy sky conditions (higher than 90%), thus limiting the direct aerosol effect. In the three other regions (EURSW, EURSE and ALPS), although
the AOD anomalies are close to zero, the effect of aerosols is stronger in winter, both in SSR (an extra dimming of about 5 W m$^{-2}$) and in temperature (an extra cooling between 0.2 and 0.4°), because of lower cloud cover in the NAO+ regime in these regions. In EURSE, the aerosol effects are even stronger despite a slight negative AOD anomaly. In summer, the relationship between AOD and the aerosol impact on SSR seems to be more linear, notably in EURSW where the positive AOD anomaly (up to 0.1) lead to an extra dimming (about 5 W m$^{-2}$). To a lesser extent, the same conclusion applies to the aerosol impact





on surface temperature, where the positive AOD anomalies cause an extra cooling of about 0.2°in EURSW and ALPS. In Northern Europe (EURNW and EURN), this regime favours days with lower cloud cover, and consequently stronger effects of aerosols for constant AOD.

To summarize, the NAO+ and AL regimes are characterized by stronger aerosol effects in Southern Europe due to different
reasons: drier conditions leading to a more efficient direct aerosol effect in winter, increase of AOD in summer. In Northern Europe, the increase in AOD due to sea-salt emissions does not generate stronger aerosol effects in winter because of important cloud cover, while in summer the decrease in cloud cover in the AL regime allows an extra dimming and cooling of aerosols.

### 5.2.3   The NAO- regime

Contrary to the previous regime, the NAO- regime is associated to a strong negative pressure anomaly over the near Atlantic,
also covering Western Europe and the Western and Central Mediterranean. Therefore storms are further south than average over the near Atlantic, and low systems are favoured over Southern Europe compared to other regimes. With regards to aerosols, sea-salt emissions are reinforced between 30 and 55°N over the Atlantic Ocean in winter (only between 40 and 55°N in summer), and weakened further north. The dust AOD pattern in winter is the opposite of the one in the NAO+ regime, with a positive anomaly in the Eastern Mediterranean and a negative anomaly off Northwestern Africa. Anthropogenic AOD anomalies are
small, and associated to precipitation anomalies. Indeed Southern Europe is concerned by a negative sulphate AOD anomaly, probably due to higher precipitation in the NAO- regime, while Northeastern Europe has a positive sulphate AOD anomaly in winter. In total, AOD anomalies consist essentially in an increase over the near Atlantic, and a decrease over Southern Europe and the Mediterranean.

In winter (Figure 14), the AOD increase over the near Atlantic leads to a decrease in clear-sky SSR by -5 to -10 W m$^{-2}$, which
is not preserved in all-sky SSR due to important cloud cover at the same place in this regime. However, the negative AOD anomalies over Southern Europe and off Northwestern Africa lead to a slight increase in clear-sky and all-sky SSR, up to 5 W m$^{-2}$ locally. No impact on surface temperature associated to these effects on SSR has been clearly identified. In summer, the negative AOD anomaly in Southern Europe leads to higher increases both in clear-sky and all-sky SSR, between 2 and 10 W m$^{-2}$ (Figure 15). However, the anomaly in the aerosol impact on surface temperature remains lower than 0.1°on average,
except in Western France where it reaches 0.2°.

More in details, the positive AOD anomaly in winter occurs simultaneously with an increase in cloud cover. Therefore, Figure 16 shows for example in EURNW less frequent days with strong aerosol effects, and more frequent days with weak aerosol effect on SSR (between 0 and -4 W m$^{-2}$). The ALPS region is concerned by the same process: more aerosols but also more clouds in this regime in winter, leading to a decrease of the aerosol impacts on SSR and surface temperature. In EURSW,
clouds are also favoured in this regime in winter, but compared to the areas further north, AOD is also reduced, leading to reduced aerosol effects. The same results are noted in summer, notably in EURNW and EURSW.

In brief, the NAO- regime is dominated by reduced aerosol effects on SSR and temperature, either due to a decrease in aerosol loads (notably in Southern Europe) or due to the reinforcement in cloud cover (for example in Western Europe) making the aerosols less efficient in their direct effect.



### 5.2.4 The Atlantic Ridge (AR) regime

As implied by its name, the AR regime is characterized by a large positive pressure anomaly over the Atlantic Ocean, which can be seen as a ridge extending from the Azores high pressure system to northern latitudes. This ridge induces a northwesterly flow over Western Europe, and is associated to low pressure anomalies over Central Europe. The pattern is similar in winter and

5 in summer, but with weaker anomalies in summer. In this regime, the aerosol loads are in most places weaker than average over Europe, for different reasons. First of all, the northwesterly flow induced by the synoptic circulation leads to more frequent precipitation in Western Europe, often under the form of showers behind cold fronts, thus scavenging the atmosphere from aerosols. Therefore, the anthropogenic AOD anomaly is negative over Europe, except in the extreme east of the domain away from this northwesterly flow. Secondly, this circulation is not favourable to dust outbreaks over the Mediterranean or even

Europe, so that the dust AOD anomaly is close to zero, or slightly negative in summer in the Western Mediterranean. Finally, the Northwestern winds generate a positive anomaly in sea-salt AOD in winter along the European coasts from the Netherlands to Spain, which is however counterbalanced by the decrease in anthropogenic AOD, except in Northern Spain.

The decrease in AOD in the AR regime leads to a positive anomaly in the aerosol impact on SSR, both in clear-sky and all-sky conditions (Figures 14 and 15). These anomalies are very weak in winter, between 0 and 2 W m$^{-2}$ in Europe, and higher in

summer, notably in clear-sky conditions (up to 10 W m$^{-2}$). The calculation of subsequent anomalies in the impact of aerosols on surface temperature shows positive anomalies in Europe larger than expected (up to 0.3°on average) given the anomalies on SSR, highlighting possible semi-direct aerosol effects.

With regards to the probability density function, Figure 16 confirms that the decrease in AOD lead to reduced aerosol impact on SSR in the five studied regions, since anomalies are positive only for dimming lower than 4 W m$^{-2}$, and cooling lower than

20 0.2°. Moreover, these AOD anomalies are associated in winter with more frequent days with important cloud cover (higher than 80%) in Western Europe (EURNW and EURSW), thus reinforcing the reduction in aerosol direct effect in this regime. In summer, cloud cover is on average weaker in most of Europe, so that the regions with negative AOD anomalies (notably EURSW, EURNW and ALPS) have more frequent days with lower aerosol effects due to lower aerosol loads. Nevertheless, in Northern Europe, the AR regime is associated with lower pressure inducing more cloud cover, thus limiting the direct aerosol

effect for unchanged AOD.

To summarize, the AR regime is unfavourable to aerosol loads over most of Europe, and their effect on SSR and temperature is thus reduced. The direct effect is even reduced in Northern Europe in summer when aerosol loads are close to average in this regime, due to increased cloud cover.

### 5.3 Synthesis

This analysis by weather regime has highlighted that aerosol loads strongly depend on the synoptic circulation, and that as a consequence, the aerosol effects on SSR and surface temperature are strongly modulated by atmospheric circulation. The role of cloud cover is essential in this modulation. As an effort to synthesize the results presented in this section, Figures 22 and 23 present a schematic map of the modulation of aerosol effects for each weather regime, respectively in winter and in summer.





On the one hand, weather regimes strongly influence aerosol loads, in all the areas delineated by the dashed lines where the plus/equal/minus symbols indicate the sign of variation. On the other hand, the modification of these aerosol loads by weather regimes has consequences on their impact on SSR and surface temperature, also depending on cloud cover. The color of the lines indicates if the aerosol impact is rather reduced (red), stable (green) or reinforced (blue) by the weather regime. The sun

and thermometer symbols indicate if this impact concerns respectively SSR or/and surface temperature. All this information has been established from the analysis provided in the previous paragraphs.

These both figures clearly show that the blocking and NAO+ regimes are mostly favourable to aerosols over the Euro-Mediterranean area, and reinforce their efficiency in their impacts on SSR and surface temperature. This is due either to a decrease of cloud cover (e.g. in Southern Europe in the NAO+ regime), or to an increase in aerosol loads (in Western Europe in

the blocking regime). However, the strong cloud cover in winter in Northern Europe in the NAO+ regime prevents an increase in aerosol radiative forcing despite higher AOD. Besides, the NAO- and Atlantic Ridge regimes result in weakening the aerosol impacts on SSR and surface temperature in most of Europe. This is the result of reduced AOD (e.g. in Southern Europe in winter) or of increased cloud fraction (e.g. in Western Europe in the winter NAO -regime). Both figures also highlight some subregional features, such as the contribution of dust aerosol effects in the summer Atlantic Low and Atlantic Ridge regimes.

**6   Conclusions**

The present study aims at better understanding climate-aerosol interactions and high-frequency aerosol variability at the synoptic scale over the Euro-Mediterranean region. The CNRM-ALADIN64 regional climate model driven by the ERA-Interim reanalysis has thus been used to better understand the interactions between the North Atlantic Oscillation, weather regimes and the different aerosol types from the interannual to daily time scales. The 40-year simulation (1979-2018) has first been evalu-

ated for various climate parameters (surface temperature, precipitation, surface wind, sea level pressure, radiation), as well as for the aerosol content against satellite and ground-based observations. Mean climate and seasonal variations are in general in good agreement between the model and observations, and significant improvements have been noted compared to the previous version of the model. The same conclusions can be drawn for the aerosols, also relevant for the aerosol daily distribution, although some discrepancies, especially the overestimation of nitrates in spring in Northern Europe, have been noted. This

model is consequently considered to be relevant for the study of climate-aerosol interactions at high temporal frequency over this region.

Two approaches have been used to explain the climate variability of aerosols, namely the NAO index and weather regimes. The first one has been shown to explain a significant part of the interannual variability, notably in winter for the export of dust aerosols over the Atlantic Ocean and the Eastern Mediterranean, and in summer for the positive anomalies of anthropogenic

aerosols over Western Europe. Nevertheless, this index is not sufficient to fully understand the variations of aerosols in this region, and their effects on regional climate. The use of weather regimes allows a better consideration of the different patterns in atmospheric circulation, which drives the emission, transport and deposition of aerosols. The issue of knowing the variations of aerosols relatively to the variations of clouds is essential to understand differences in aerosol effects on shortwave surface



radiation and surface temperature.

The four weather regimes usually defined in this area in winter and in summer bring significant information to answer this question. Two synthesis figures, namely Figures 22 and 23, have been established to summary the modulation of aerosol effects on surface radiation and surface temperature as a function of weather regimes. In the blocking regime, aerosols have
been shown to be more efficient in their interactions with radiation, either because cloud cover is less important (in Northern Europe), or because of higher concentrations due to more stable conditions (in Southern Europe). In Southern Europe, aerosol impacts on climate are also stronger in the NAO+ and Atlantic Low regimes due to drier conditions in winter, and due to higher loads in summer. On the contrary, the strong cloud cover in winter in Northern Europe prevents an increase in aerosol radiative forcing despite higher AOD. In the NAO- regime, aerosols are less efficient in their direct effect, either due to reduced AOD
(e.g. in Southern Europe) or due to the reinforcement in cloud cover (e.g. in Western Europe). Finally, the AR regime is also unfavourable to aerosol radiative effects, since aerosol loads are generally weaker in this regime, and cloud cover is also higher in Northern Europe.

As a matter of fact, this study highlights the need of considering high-frequency aerosol variations to better represent climate-aerosol interactions, and therefore regional climate itself. This kind of processes cannot been properly assessed in regional
climate models using monthly aerosol optical depth climatologies. We could presume that a regional climate model with only monthly AOD climatology, as many of them exist in the Euro-CORDEX and Med-CORDEX programs for instance, could underestimate or overestimate the effects of aerosols in several weather regimes. Since cloud and aerosol variations are not uncorrelated, the frequency of clear-sky days with averaged and relatively high AOD could be for example overestimated in case of using monthly AOD dataset.

*Code and data availability.*   The code of the regional climate model CNRM-ALADIN64 is available as follows: the SURFEX code is accessible using a CECILL-C Licence (http://www.cecill.info/licences/Licence_CeCILL-C_V1-en.txt) at http://www.umr-cnrm.fr/surfex; OASIS3-MCT is available at https://verc.enes.org/oasis/download; XIOS at https://forge.ipsl.jussieu.fr/ioserver and the rest of the ALADIN-Climat code is available upon request to the authors. The two climate simulations used in this study are also available by contacting the authors. All the other datasets used in this study (satellite and ground-based observation) are available at the different websites mentioned in Section **??**.

*Competing interests.*   No competing interests are present.

*Acknowledgements.*   We thank the support of the whole team in charge of the CNRM climate models, and especially that of Antoinette Alias and Stéphane Sénési for their technical assistance. Supercomputing time and financial support of this publication was provided by Météo-France. We acknowledge all the PI investigators of the AERONET stations and their staff for establishing and maintaining the 72 sites used in the present work. We are also grateful to the BSRN network for providing radiation data in the 10 stations used here. We would
like also to thank the NASA Langley Research Center Atmospheric Science Data Center, the National Center for Atmospheric Research



and the Climate Research Unit for providing the data sets used to evaluate our climate simulations. This work is part of the Med-CORDEX initiative (www.medcordex.eu) and a contribution to the ChArMEx programm, part of the French multidisciplinary program MISTRALS (Mediterranean Integrated Studies aT Regional And Local Scales).



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





| Atmosphere (ALADIN-Climat) | |
|---|---|
| Aerosol optical properties | Nabat et al. (2013) |
| Cloud optical properties | Liquid (Slingo, 1989) and ice (Fu, 1996) clouds |
| Cloud scheme | Sommeria and Deardorff (1977); Bougeault (1981); Ricard and Royer (1993) |
| Convection | PCMT (Piriou et al., 2007; Guérémy, 2011) |
| Gravity wave drag | Orographic (Déqué et al., 1994; Catry et al., 2008) and non-orographic (Lott and Miller, 1997) |
| Indirect aerosol effect | Cloud-albedo effect following Menon et al. (2002) |
| Microphysics | Lopez (2002) |
| Radiative transfer | Longwave (RRTM, Mlawer et al., 1997) and |
| | shortwave (FMR with 6 bands, Fouquart and Bonnel, 1980; Morcrette et al., 2008) |
| Turbulence | Cuxart et al. (2000) |

| Surface (SURFEX) | |
|---|---|
| Lakes | FLake (Le Moigne et al., 2016) |
| Land-surface | ISBA (Decharme et al., 2019) |
| River routine model | CTRIP (Decharme et al., 2019) |
| Sea-surface fluxes | ECUME (Belamari and Pirani, 2007) |

| Aerosol scheme (TACTIC) | |
|---|---|
| Conversion $SO_2$-$SO_4$ | Huneeus (2007) |
| Dry deposition | Reddy et al. (2005); Morcrette et al. (2009) |
| Dust emission | Marticorena and Bergametti (1995); Kok (2011b) |
| Nitrate-Ammonium formation | Hauglustaine et al. (2014) |
| Sea-salt emission | Grythe et al. (2014) |
| Sedimentation | Tompkins et al. (2005) |
| Wet deposition | In-cloud (Giorgi and Chameides, 1986) and below-cloud scavenging Morcrette et al. (2009) |

**Table 1.** Summary of the parameterizations used in CNRM-ALADIN64, for atmosphere (ALADIN-Climat), surface (SURFEX) and aerosol scheme (TACTIC).





Table 2: List of AERONET and BSRN stations used in this study. Latitudes, longitudes, elevations and periods of time are given for each station. Stations are gathered in the 9 regions (A to I) defined in Figure 1.

| Region | # | Station | Lat | Lon | Elevation | Time | Network |
|--------|-----|---------|------|-------|-----------|------|---------|
| | 1 | Edinburgh | 55.9 | -3.2 | 97.5 | 2011-2015 and 2017-2018 | AERONET |
| | 2 | Chilbolton | 51.1 | -1.4 | 88.0 | 2005-2017 | AERONET |
| | 3 | Rame_Head | 50.3 | -4.2 | 105.0 | 1997-1998 and 2013-2018 | AERONET |
| | cam | Camborne | 50.2 | -5.3 | 88 | 2001-2016 | BSRN |
| A | 4 | Arcachon | 44.7 | -1.2 | 11.0 | 2008-2017 | AERONET |
| | 5 | Coruna | 43.4 | -8.4 | 67.0 | 2012-2018 | AERONET |
| | 6 | Evora | 38.6 | -7.9 | 293.0 | 2003-2018 | AERONET |
| | 7 | Cabo_da_Roca | 38.8 | -9.5 | 136.0 | 2003-2008 and 2010-2018 | AERONET |
| | 8 | Azores | 38.5 | -28.6 | 50.0 | 2000-2004 | AERONET |
| | 9 | Capo_Verde | 16.7 | -22.9 | 60.0 | 1999-2018 | AERONET |
| | 10 | Dakar | 14.4 | -17.0 | 21.0 | 1996-1997 and 2000-2016 | AERONET |
| | 11 | Dahkla | 23.7 | -15.9 | 12.0 | 2002-2005 | AERONET |
| | 12 | Santa_Cruz_Tenerife | 28.5 | -16.2 | 52.0 | 2005-2018 | AERONET |
| B | 13 | Saada | 31.6 | -8.2 | 420.0 | 2004-2017 | AERONET |
| | 14 | Ouarzazate | 30.9 | -6.9 | 1136.0 | 2012-2015 | AERONET |
| | 15 | Blida | 36.5 | 2.9 | 230.0 | 2003-2010 and 2012 | AERONET |
| | 16 | Tamanrasset_INM | 22.8 | 5.5 | 1377.0 | 2006-2018 | AERONET |
| | tam | Tamanrasset | 22.8 | 5.5 | 1385 | 2000-2016 | BSRN |
| | 17 | El_Farafra | 27.1 | 28.0 | 92.0 | 2014-2018 | AERONET |
| | 18 | Cairo_EMA_2 | 30.1 | 31.3 | 70.0 | 2010-2017 | AERONET |
| | 19 | SEDE_BOKER | 30.9 | 34.8 | 480.0 | 1997-2018 | AERONET |
| | sbo | Sede Boker | 30.9 | 34.8 | 500 | 2003-2012 | BSRN |
| | 20 | Eilat | 29.5 | 34.9 | 15.0 | 2007-2009 and 2011-2018 | AERONET |
| C | 21 | KAUST_Campus | 22.3 | 39.1 | 11.2 | 2012-2016 | AERONET |
| | 22 | Solar_Village | 24.9 | 46.4 | 764.0 | 1999-2013 | AERONET |
| | sov | Solar_Village | 24.9 | 46.4 | 650 | 1998-2002 | BSRN |
| | 23 | Kuwait_University | 29.3 | 48.0 | 42.0 | 2007-2010 | AERONET |
| | 24 | IASBS | 36.7 | 48.5 | 1805.0 | 2009-2013 and 2016-2018 | AERONET |
| | 25 | Malaga | 36.7 | -4.5 | 56.0 | 2008-2016 | AERONET |
| | 26 | Tabernas_PSA-DLR | 37.1 | -2.4 | 500.0 | 2011-2012 and 2014-2018 | AERONET |
| | 27 | Burjassot | 39.5 | -0.4 | 104.0 | 2007-2018 | AERONET |
| D | | | | | | | |





| Region | # | Station | Lat | Lon | Elevation | Time | Network |
|--------|---|---------|-----|-----|-----------|------|---------|
| | 28 | Palma_de_Mallorca | 39.6 | 2.6 | 10.0 | 2011-2018 | AERONET |
| | 29 | Barcelona | 41.4 | 2.1 | 125.0 | 2004-2018 | AERONET |
| | 30 | Montsec | 42.1 | 0.7 | 1574.0 | 2011-2018 | AERONET |
| | 31 | Zaragoza | 41.6 | -0.9 | 250.0 | 2012-2018 | AERONET |
| | cnr | Cener | 42.8 | -1.6 | 471 | 2009-2016 | BSRN |
| | 32 | Palencia | 42.0 | -4.5 | 750.0 | 2003-2017 | AERONET |
| | 33 | Tunis_Carthage | 36.8 | 10.2 | 10.0 | 2013-2017 | AERONET |
| | 34 | Lampedusa | 35.5 | 12.6 | 45.0 | 2000-2006 and 2010-2018 | AERONET |
| | 35 | IMC_Oristano | 39.9 | 8.5 | 10.0 | 2000-2003 | AERONET |
| E | 36 | Messina | 38.2 | 15.6 | 15.0 | 2005-2018 | AERONET |
| | 37 | Lecce_University | 40.3 | 18.1 | 30.0 | 2003-2016 | AERONET |
| | 38 | IMAA_Potenza | 40.6 | 15.7 | 770.0 | 2005-2018 | AERONET |
| | 39 | Rome_Tor_Vergata | 41.8 | 12.6 | 130.0 | 2001-2017 | AERONET |
| | 40 | Ersa | 43.0 | 9.4 | 80.0 | 2008-2018 | AERONET |
| | 41 | Nes_Ziona | 31.9 | 34.8 | 40.0 | 2000-2014 | AERONET |
| | 42 | FORTH_CRETE | 35.3 | 25.3 | 20.0 | 2003-2017 | AERONET |
| | 43 | CUT-TEPAK | 34.7 | 33.0 | 22.0 | 2010-2012 and 2014-2018 | AERONET |
| F | 44 | IMS-METU-ERDEMLI | 36.6 | 34.3 | 3.0 | 1999-2001 and 2003-2018 | AERONET |
| | 45 | ATHENS-NOA | 38.0 | 23.7 | 130.0 | 2008-2017 | AERONET |
| | 46 | Thessaloniki | 40.6 | 23.0 | 60.0 | 2003-2018 | AERONET |
| | 47 | Xanthi | 41.1 | 24.9 | 54.0 | 2008-2015 | AERONET |
| | 48 | TUBITAK_UZAY_Ankara | 39.9 | 32.8 | 924.0 | 2009-2012 and 2017 | AERONET |
| | 49 | Carpentras | 44.1 | 5.1 | 107.0 | 2003-2015 | AERONET |
| | car | Carpentras | 44.1 | 5.1 | 100 | 1996-2016 | BSRN |
| | 50 | Villefranche | 43.7 | 7.3 | 130.0 | 2004-2008 and 2010-2016 | AERONET |
| | 51 | Modena | 44.6 | 10.9 | 56.0 | 2000-2016 | AERONET |
| | 52 | Venise | 45.3 | 12.5 | 10.0 | 2001-2011 | AERONET |
| G | 53 | Davos | 46.8 | 9.8 | 1589.0 | 2001 and 2004-2018 | AERONET |
| | 54 | Ispra | 45.8 | 8.6 | 235.0 | 1997-2010 | AERONET |
| | pay | Payerne | 46.8 | 6.9 | 491 | 1996-2011 | BSRN |
| | 55 | Munich_University | 48.1 | 11.6 | 533.0 | 2001-2002 and 2007-2017 | AERONET |
| | 56 | Palaiseau | 48.7 | 2.2 | 156.0 | 1999-2000 and 2002-2018 | AERONET |
| | pal | Palaiseau | 48.7 | 2.2 | 156 | 2005-2016 | BSRN |
| | 57 | Belsk | 51.8 | 20.8 | 190.0 | 2002-2016 | AERONET |

H





| Region | # | Station | Lat | Lon | Elevation | Time | Network |
|---|---|---|---|---|---|---|---|
| | 58 | CLUJ_UBB | 46.8 | 23.6 | 405.0 | 2010-2018 | AERONET |
| | 59 | Bucharest_Inoe | 44.3 | 26.0 | 89.0 | 2007-2016 | AERONET |
| | 60 | Moldova | 47.0 | 28.8 | 205.0 | 1999-2018 | AERONET |
| | 61 | Sevastopol | 44.6 | 33.5 | 80.0 | 2006-2013 | AERONET |
| | 62 | Kyiv | 50.4 | 30.5 | 200.0 | 2007-2018 | AERONET |
| | 63 | Minsk | 53.9 | 27.6 | 235.0 | 2003-2018 | AERONET |
| | 64 | Moscow_MSU_MO | 55.7 | 37.5 | 192.0 | 2001-2018 | AERONET |
| | 65 | Dunkerque | 51.0 | 2.4 | 5.0 | 2003-2018 | AERONET |
| | 66 | Cabauw | 52.0 | 4.9 | 0 | 2003 and 2007-2017 | AERONET |
| | cab | Cabauw | 52.0 | 4.9 | 0 | 2005-2016 | BSRN |
| | 67 | Helgoland | 54.2 | 7.9 | 33.0 | 2000-2014 | AERONET |
| | 68 | Hamburg | 53.6 | 10.0 | 120.0 | 2000 and 2002-2018 | AERONET |
| I | 69 | Mainz | 50.0 | 8.3 | 150.0 | 2003-2018 | AERONET |
| | 70 | Leipzig | 51.4 | 12.4 | 125.0 | 2001-2018 | AERONET |
| | lin | Lindenberg | 52.2 | 14.1 | 125 | 1996-2016 | BSRN |
| | 71 | Gotland | 57.9 | 19.0 | 10.0 | 1999-2004 | AERONET |
| | 72 | Birkenes | 58.4 | 8.3 | 230.0 | 2009-2018 | AERONET |





| DJF | Temperature | | Precipitation | | Cld | SLP | JJA | Temperature | | Precipitation | | Cld | SLP |
|---|---|---|---|---|---|---|---|---|---|---|---|---|---|
| Region | ALD | ECx | ALD | ECx | ALD | ALD | Region | ALD | ECx | ALD | ECx | ALD | ALD |
| ME | -0.1 | -1.4/0.7 | 35 | -17/60 | 13 | -0.2 | ME | 0.6 | -1.4/0.8 | -19 | -50/52 | -8 | -0.1 |
| EA | -0.2 | -1.8/0.6 | 37 | -16/74 | 10 | -0.3 | EA | 1.8 | -0.8/1.6 | -34 | -33/64 | -10 | -0.7 |
| FR | -0.3 | -1.7/0.7 | 28 | -9/75 | 10 | 0.1 | FR | 0.8 | -1.4/0.9 | -28 | -47/63 | -7 | -0.3 |
| AL | -1.3 | -3.9/0.2 | 18 | -5/57 | 10 | 0.4 | AL | 1.3 | -2.4/1.0 | -29 | -39/50 | 3 | 0.2 |
| IP | -1.0 | -2.2/0.4 | 32 | -15/44 | 6 | 0.6 | IP | 1.1 | -1.8/1.3 | -37 | -47/75 | -4 | -0.5 |
| MD | -0.1 | -2.8/0.3 | 28 | 4/66 | -2 | 0.7 | MD | 1.4 | -1.6/2.4 | -50 | -39/182 | -4 | -0.1 |

**Table 3.** Averaged biases in ALD-AER simulation (ALD) calculated over the PRUDENCE boxes (shown in Figure 2), in terms of surface temperature (in °C), precipitation (in %), cloud cover (Cld in %) and sea level pressure (SLP in hPa) for winter (DJF) and summer (JJA). Corresponding biases for Euro-CORDEX (ECx) models (Kotlarski et al., 2014) have been added for temperatature and precipitation.



**Table 4.** DJF (left) and JJA (right) correlations between AOD anomalies (for ALADIN, MODIS and MISR) and NAO index over six regions (AFRW, EURSE, EURNW, EURSW, ALPS and EURN) whose limits are given in this table (also shown in Fig. 9). Significant values at the level 0.05 are noted in bold font.

| Region | Limits | ALADIN | ALADIN | MODIS | MISR | ALADIN | ALADIN | MODIS | MISR |
|---|---|---|---|---|---|---|---|---|---|
| | | | DJF | | | | JJA | | |
| | | 1980-2017 | | 2001-2017 | | 1979-2017 | | 2001-2017 | |
| AFRW | 15-27°N, 0-23°W | **0.77** | **0.82** | **0.75** | **0.76** | 0.08 | 0.19 | -0.11 | 0.12 |
| EURSE | 36-44°N, 20-27°E | -0.15 | -0.20 | -0.35 | **-0.61** | **0.33** | 0.15 | 0.19 | 0.12 |
| EURNW | 46-55°N, 10°W-2°E | **0.44** | **0.52** | 0.26 | 0.19 | 0.27 | 0.11 | 0.27 | **0.57** |
| EURSW | 37-44°N, 5°W-6°E | 0.09 | 0.43 | 0.03 | 0.05 | 0.32 | 0.28 | 0.25 | 0.15 |
| ALPS | 44-48°N, 7-15°E | -0.19 | 0.14 | **-0.54** | -0.46 | **0.34** | 0.00 | 0.32 | 0.03 |
| EURN | 7-20°N, 50-57°E | 0.31 | **0.55** | -0.41 | -0.26 | -0.04 | -0.06 | 0.14 | 0.17 |







**Figure 1.** Domain and orography (m) used in ALADIN-Climat simulations. AERONET and BSRN stations used in this study have been added with coloured crosses and circles respectively (See Table 2), as well as the nine subregions (A to I) in which they are gathered.



## a) Sea level pressure

DJF                                        ALADIN - ERAI

JJA                                        ALADIN - ERAI

EA
ME
FR
AL
IP            MD

-3    -2    -1    0    1    2    3    hPa

## b) Surface wind

DJF                                    ALADIN - QuikSCAT

DJF                                    ALADIN - QuikSCAT

-10  -8   -6   -4   -2   0    2    4    6    8   10    km/h

**Figure 2.** Winter (DJF, left) and summer (JJA, right) average differences between ALADIN and ERA-Interim for sea level pressure (hPa, 1979-2016, top), and between ALADIN and QuikSCAT for surface wind (km/h, 2000-2009, bottom). One wind barb represents 2 km/h.





**Figure 3.** Winter (DJF, left) and summer (JJA, right) average differences between ALADIN and observations (CRU and Cloud-SAT/CALIPSO) for 2m-temperature (°C, 1979-2015, a), precipitation (mm/day, 1979-2015 b) and cloud cover (%, 2006-2011, total fraction in c, low fraction in d).





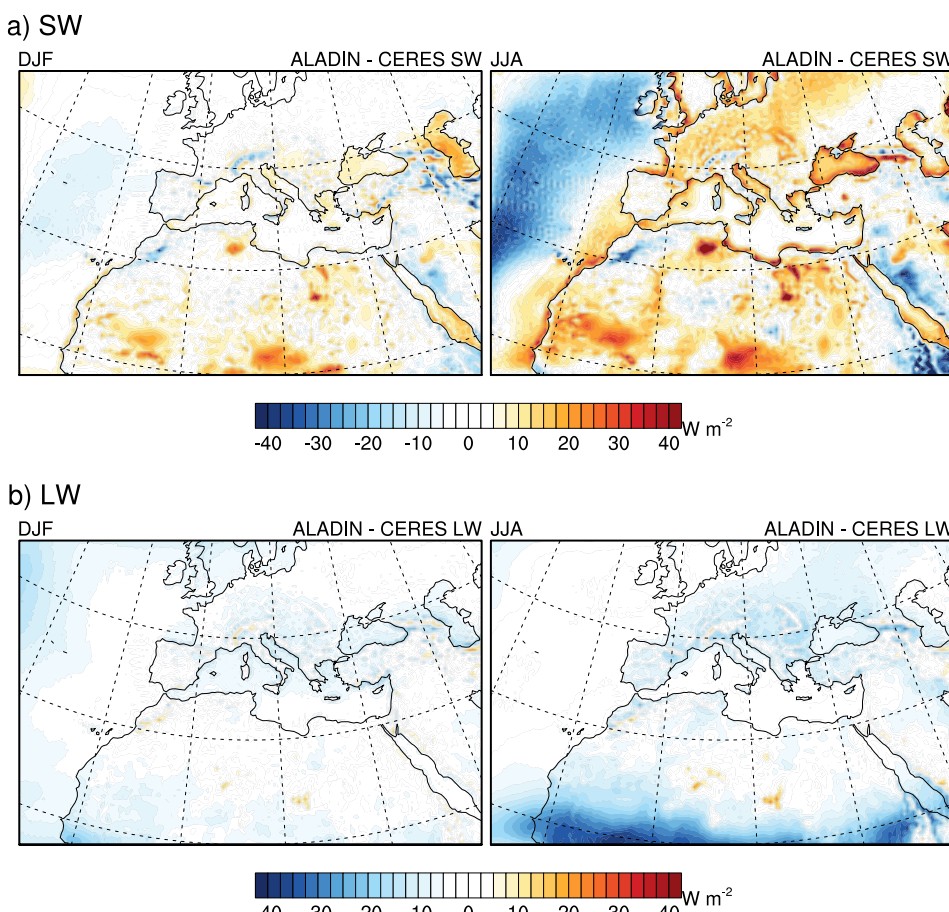

**Figure 4.** Winter (DJF, left) and summer (JJA, right) average differences between ALADIN and CERES (2000-2016) for SW (a) and LW (b) TOA radiation (W m$^{-2}$).

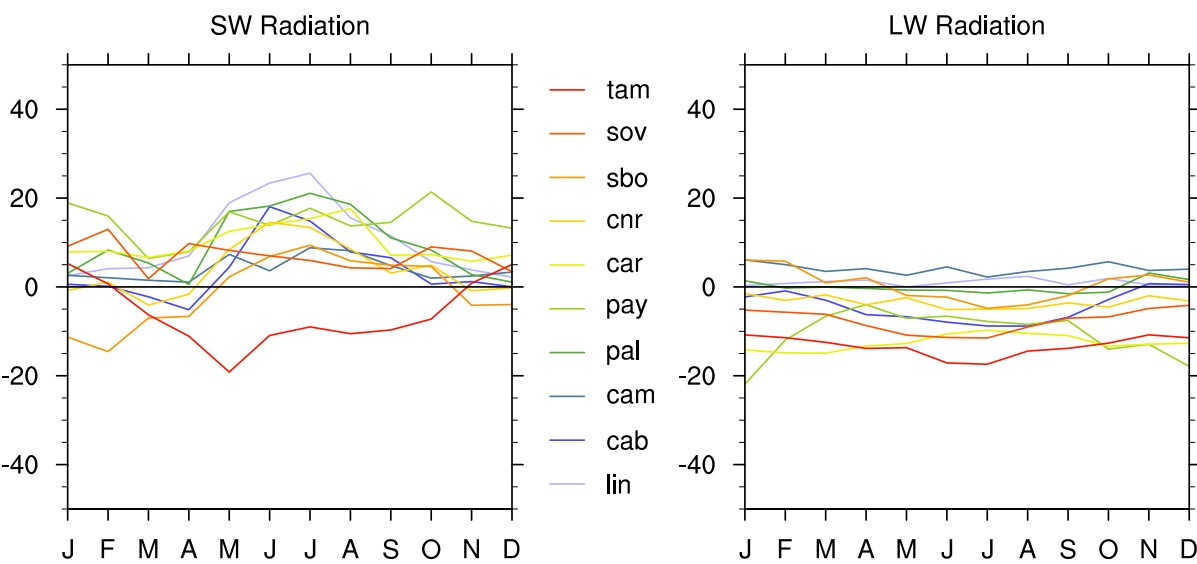

**Figure 5.** Monthly average differences between ALADIN and surface downwelling radiation fluxes (shortwave on the left, longwave on the right, in $W\,m^{-2}$) from 10 BSRN stations (presented in Table 2 and in Figure 1). The period used is noted in Table 2 for each station.





**Figure 6.** Averaged AOD (total and for each aerosol type) at 550 nm simulated by ALADIN-Climat between 2003 and 2017. Total AOD from MODIS and MISR has been included on the first line.

**Figure 7.** Monthly means in AERONET stations (black lines) and ALADIN simulation (color bars) for aerosol optical depth at 550 nm. The contribution of each aerosol type in ALADIN simulation is given by the following color bars : dark blue for sea-salt, brown for dust, red for sulphate, green for organic matter, yellow for black carbon, purple for nitrates and light blue for ammonium. Only the days where observations are available have been taken into account in the model.





**Figure 8.** Box plots comparing daily AOD simulated by ALADIN-Climat (in black) to AERONET measurements (in grey). The numbers in the x-axis correspond to the respective stations defined in Figure 1 and Table 2. The limits of each boxplot are given by the first and third quartiles, the inner horizontal line is the median, while the whiskers limited by the 5th and 95th percentiles. Colored crosses represent the average total AOD (AERONET in grey, ALADIN-Climat in black) and the contribution of each aerosol type in ALADIN-Climat (dust in brown, sea-salt in dark blue, sulphate in red, organic matter in green, black carbon in yellow, nitrates in purple and ammonium in light blue). Only the days where observations are available have been taken into account in the model.



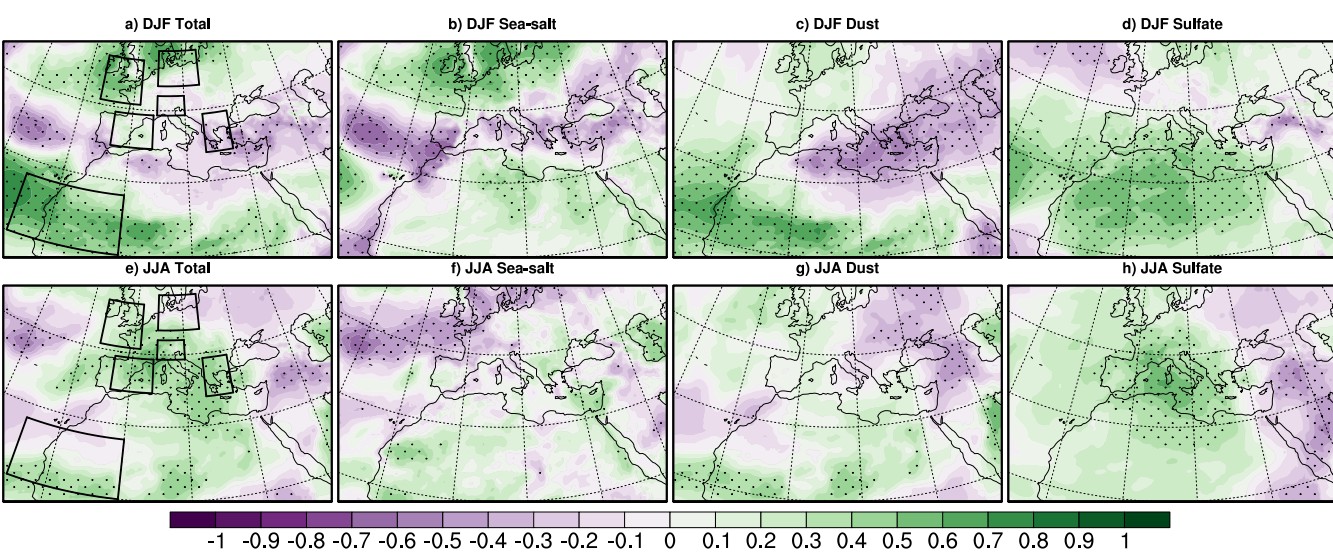

**Figure 9.** Correlation between NAO index and AOD (total and by aerosol type) at 550nm simulated by ALADIN-Climat over the period 1979-2018, separately for DJF (a-d) and JJA (e-h). Dotted areas are significant at the level 0.05



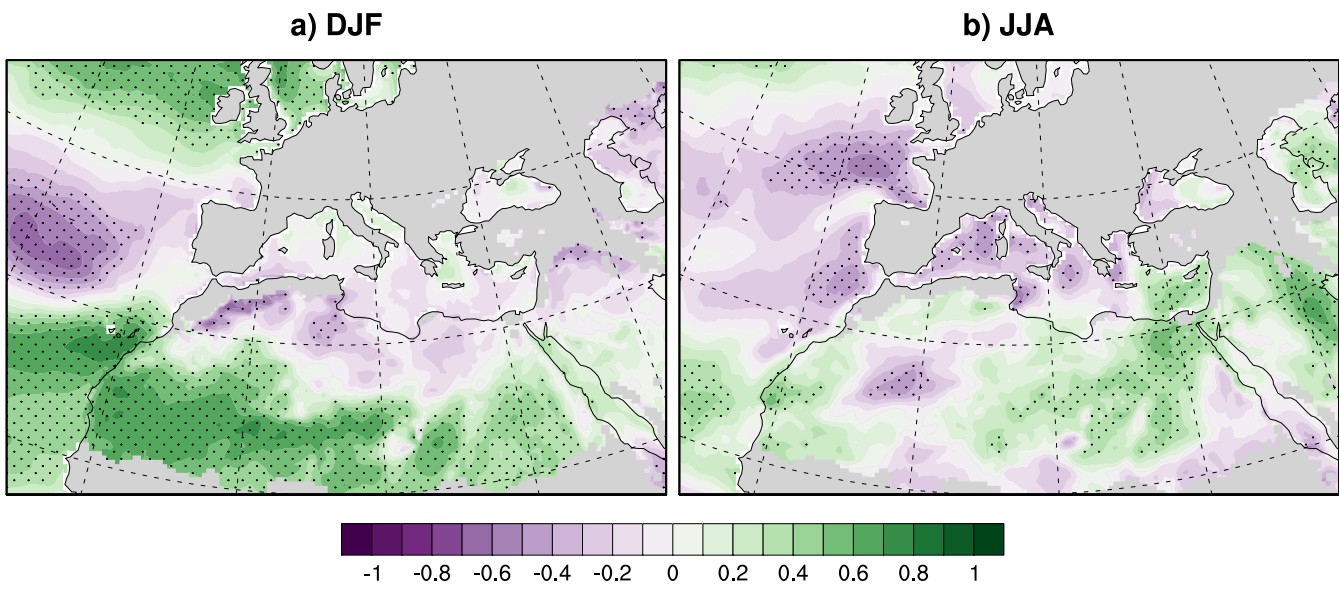

**Figure 10.** Correlation between NAO index and natural emissions (dust emissions over land, sea-salt emissions over ocean) simulated by ALADIN-Climat over the period 1979-2018, for DJF (a) and JJA (b). Grey color shows areas without dust and sea-salt emissions. Dotted areas are significant at the level 0.05

**Figure 11.** Winter (DJF, left) and summer (JJA, right) means of NAO index (bottom) and AOD anomalies (in %, first three lines) in three regions (EURNW, EURSE and AFRW). AOD simulated data are in color bars while AOD satellite data are in colored circles (MODIS in purple, MISR in green). AOD anomalies are calculated against the common 2001-2017 period.

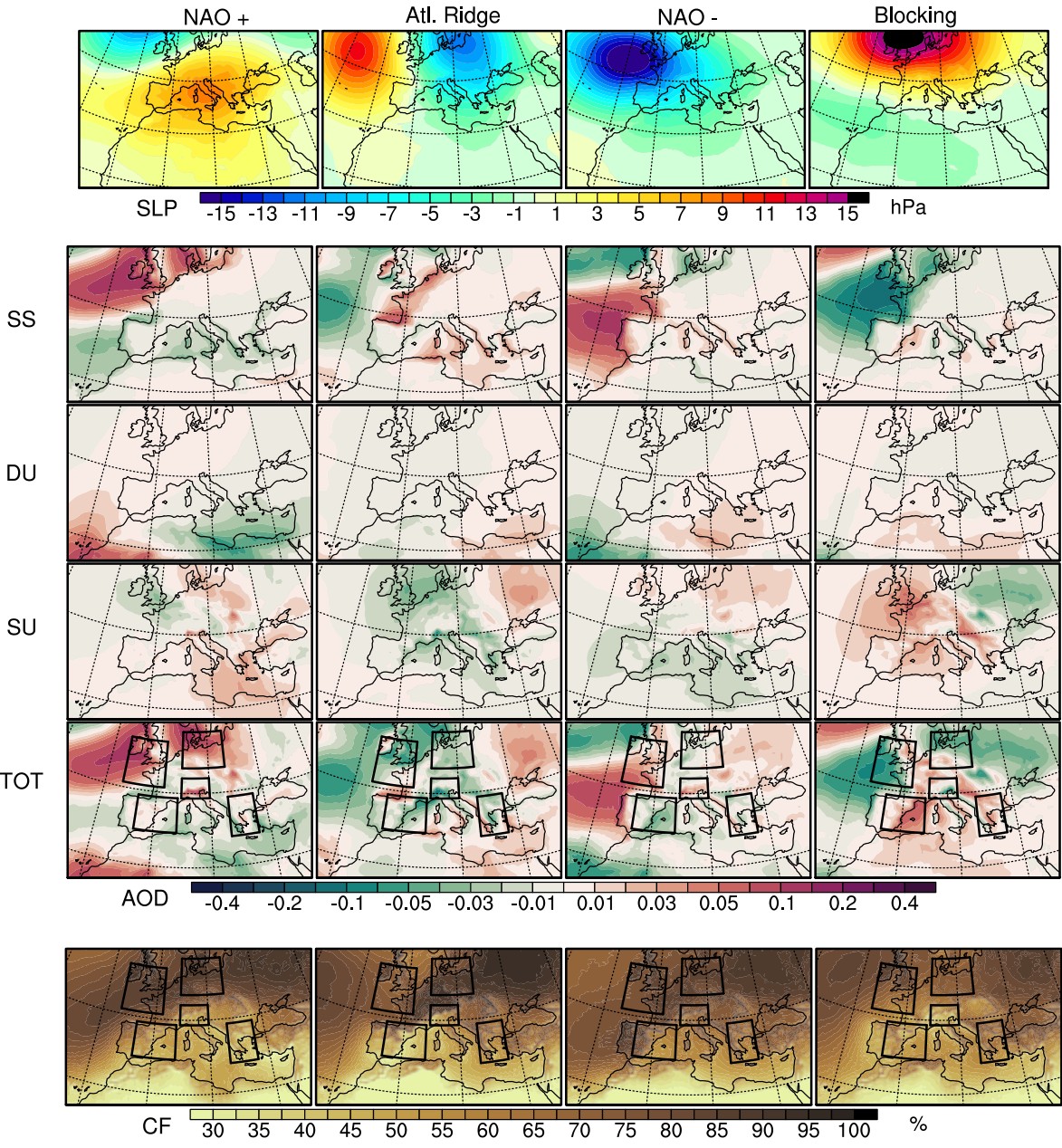

**Figure 12.** Winter (DJF) anomalies in sea level pressure (in hPa, first line), AOD at 550 nm (SS for sea-salt, DU for dust, SU for sulphate and TOT for total), and cloud cover (in %, last line) for each weather regime (NAO+, Atlantic Ridge, NAO- and blocking).





**Figure 13.** Summer (JJA) anomalies in sea level pressure (in hPa, first line), AOD at 550 nm (SS for sea-salt, DU for dust, SU for sulphate and TOT for total), and cloud cover (in %, last line) for each weather regime (NAO-, Atlantic Ridge, Atlantic Low and blocking).





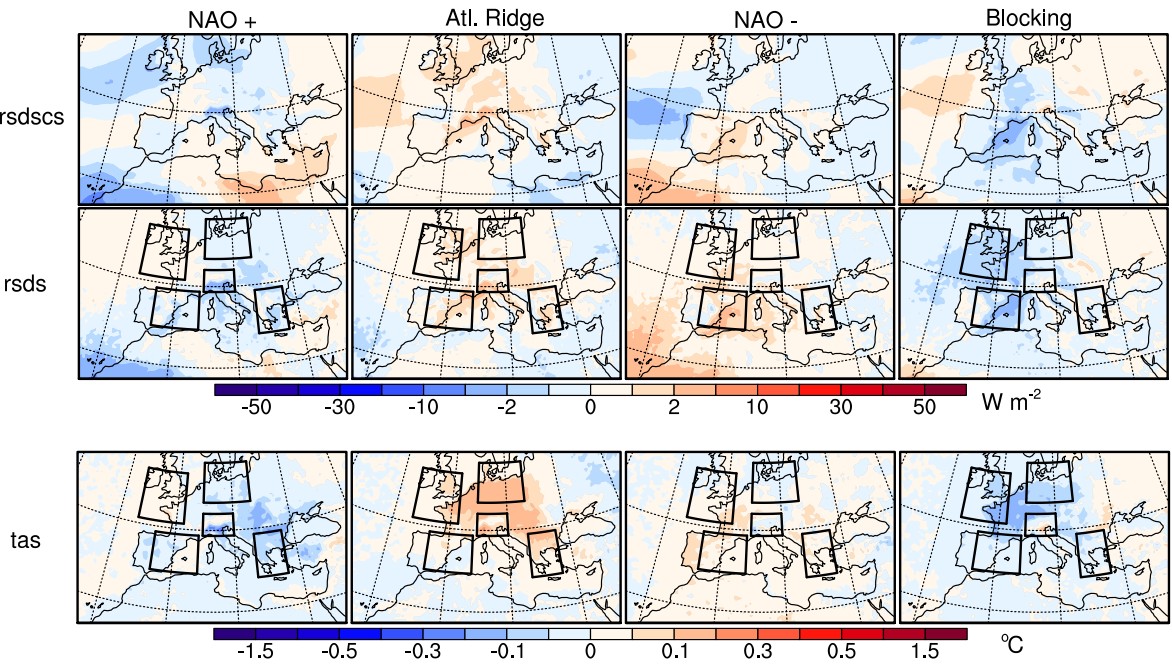

**Figure 14.** Winter (DJF) anomalies in the impact of aerosols (ALD-AER - ALD-NO) on surface SW downward radiation (rsdscs for clearsky and rsds for all-sky, in W m$^{-2}$) and on surface temperature (tas in °C) for each weather regime (NAO+, Atlantic Ridge, NAO- and blocking).



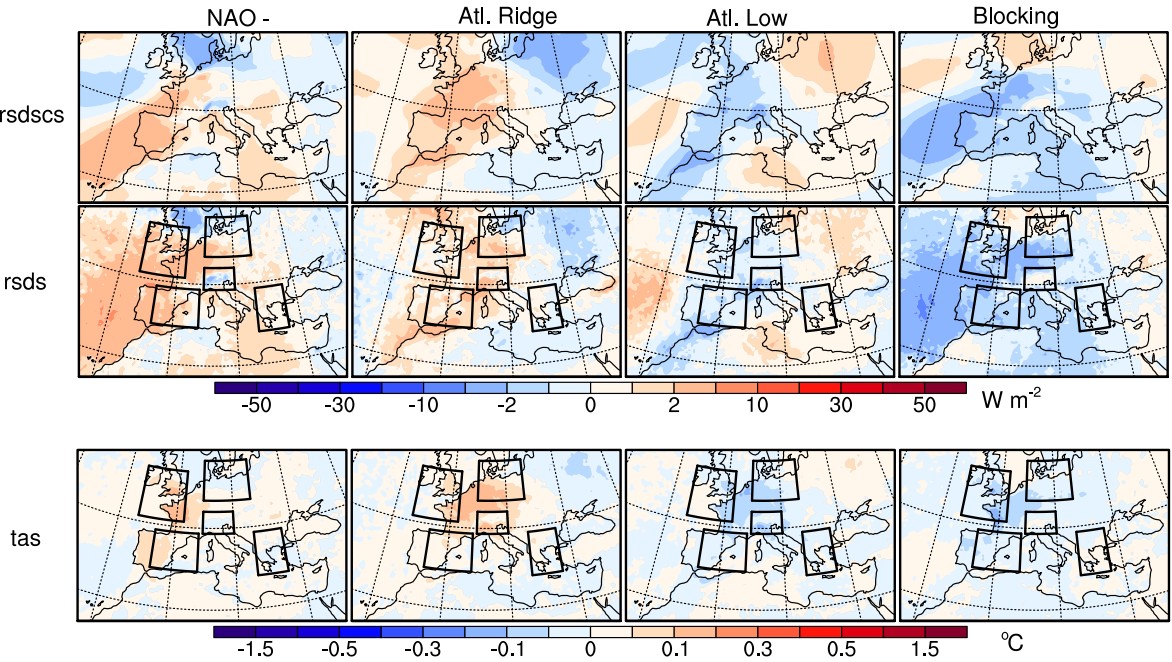

**Figure 15.** Summer (JJA) anomalies in the impact of aerosols (ALD-AER - ALD-NO) on surface SW downward radiation (rsdscs for clear-sky and rsds for all-sky, in W m$^{-2}$) and on surface temperature (tas in °C) for each weather regime (NAO-, Atlantic Ridge, Atlantic Low and blocking).





**Figure 16.** Probability distribution functions of the aerosol impact on surface SW downward radiation (in W m$^{-2}$, calculated as the difference between ALD-AER and ALD-NO) against AOD at 550 nm in winter (DJF). The black lines represent the average, while the colours present the anomalies of this distribution for each weather regime.



**Figure 17.** Probability distribution functions of cloud cover (in %) against AOD at 550 nm in winter (DJF). The black lines represent the average, while the colours present the anomalies of this distribution for each weather regime.





**Figure 18.** Probability distribution functions of the aerosol impact on surface temperature (in °C, calculated as the difference between ALD-AER and ALD-NO) against AOD at 550 nm in winter (DJF). The black lines represent the average, while the colours present the anomalies of this distribution for each weather regime.





**Figure 19.** Probability distribution functions of the aerosol impact on surface SW downward radiation (in W m$^{-2}$, calculated as the difference between ALD-AER and ALD-NO) against AOD at 550 nm in summer (JJA). The black lines represent the average, while the colours present the anomalies of this distribution for each weather regime.

**Figure 20.** Probability distribution functions of cloud cover (in %) against AOD at 550 nm in summer (JJA). The black lines represent the average, while the colours present the anomalies of this distribution for each weather regime.





**Figure 21.** Probability distribution functions of the aerosol impact on surface temperature (in °C, calculated as the difference between ALD-AER and ALD-NO) against AOD at 550 nm in summer (JJA). The black lines represent the average, while the colours present the anomalies of this distribution for each weather regime.



**Figure 22.** Synthesis scheme of the aerosol effects as a function of weather regimes in winter (DJF). The main anomalies in aerosol optical depth and aerosol impact on surface radiation and temperature for each weather regime are summarized by areas delineated by dashed lines. Red colour refers to a reduced impact of aerosols, green to similar impact, and blue to a reinforced impact of aerosols. The colored plus/equal/minus symbol indicates the AOD anomaly, while the plus/minus symbol inside clouds indicate the cloud cover anomaly. The inclusion of sun/thermometer symbols indicate a respective impact of aerosols on surface radiation/surface temperature in the subregion. Grey colors show the average cloud cover fraction for each weather regime.







**Figure 23.** Same as Figure 22 but for summer (JJA).