# Peer review of "Modulation of radiative aerosols effects by atmospheric circulation over the Euro-Mediterranean region"

_Atmospheric Chemistry and Physics, 2019_

## Referee Comment (RC1) · Anonymous Referee #1 · 25 Feb 2020

This manuscript investigates the links between aerosols variability modeled by CNRM-ALADIN64 regional climate model with and synoptic atmospheric circulation over the Euro-Mediterranean region, including analysis with respect to the variability of North-Atlantic Oscillation as well as to weather regimes based on persisting meteorological patterns. It is well structured and written and illustrates original and interesting results. I suggest acceptance of the manuscript for publication after taking into consideration the following comments.

Comments 1) Page 2, lines 15-16: The authors may also consider that Mediterranean cyclones developing in winter and autumn could also affect the dust transport at the

Eastern Mediterranean (Flaounas et al., 2015; Georgoulias et al., 2016). 2) Page 2, line 18: Please, add a relevant reference 3) Page 2, line 19: add a reference. This shift of dust load from Eastern Mediterranean in spring to western Mediterranean in summer has been also shown in a recent study by Marinou et al. (2017) using a satellite pure dust product based on CALIPSO. 4) Page 10, lines 0-10: I would suggest to discuss shortly the biases in a quantitative manner with respect to MODIS and MISR. Maybe the authors could think of adding a field of the biases in Figure 6, but this is optional. Of course, in the following section, the authors discuss the AOD biases with respect to station data. 5) Page 12, lines 2-3: The NAO index used for the analysis would be more consistent if it would be based on ERA-interim which drives ALD-AER rather than the NAO-index provided by NOAA. Could the authors comment on this issue and justify the use of NOAA instead calculated from ERA-interim? 6) Page 12, lines 16-18: The justification provided by the authors is very reasonable. I think that a strong support on all this discussion would be provided by plotting near surface wind vectors along with model AOD separately for the positive phase and the negative NAO phase. This is, however, a suggestion which could be only optionally considered by the authors. 7) Page 13, lines 19-20: Here it is mentioned that less rainfall is noted in summer under the positive phase of NAO but according to Blade et al. (2011) during high NAO summers, when strong anticyclonic conditions and suppressed precipitation prevail over the UK, the Mediterranean region instead is anomalously wet (see for example their Figure 10). Could you check in your analysis the regions that less rainfall is noted in summer under the positive phase of NAO? 8) Page 20, lines 15-18: Could you make a course estimate of the impact of using monthly instead of daily AOD climatology in RCM simulations? I guess this can have a stronger impact on SSR but the impact on near surface temperature is maybe trivial. 9) Figure 1: Please describe in figure caption the numbering. 10) Figure 2: QuikSCAT is presented here but not described. Please add in Section 2 a description of QuikSCAT dataset used here for model evaluation of sea winds. Discuss also how the spatial resolution of the model and observation data compares. You may also think also of presenting a comparison

of ALADIN surface wind with ERA-Interim for consistency with the SLP comparison. 11) Figure 3: The low cloud and total cloud differences plot is of pure quality and not really informative because of the gaps. Discuss also how the spatial resolution of the model and observation data compares.

Minor Corrections 1) Page 2, line 9: Please, specify which season. 5) Page 12, lines 17: easterly winds instead of eastern winds.

---

## Referee Comment (RC2) · Anonymous Referee #2 · 19 Mar 2020

Nabat et al. present firstly a new model version of a regional climate model that has a number of revised parameterisations compared to a previous version. An evaluation with multiple observational datasets is presented. This evaluation is a bit hampered by the fact that no direct comparison is presented between the new model version and an older one, even if often the text compares the skill of the previous version (unknown to the reader unless they carefully studied the former papers by the authors). In particular the aerosol distributions and temporal variability are compared to satellite retrievals and surface remote sensing. The models shows a rather remarkable skill both for the geographical distribution and the annual cycles of aerosol optical depths. The bulk of the manuscript is a lengthy analysis on how aerosols are simulated differently

for different weather conditions as firstly defined by the NAO index (presumably as seasonal averages, the text needs to be clarified on the temporal resolution of the analysis), and secondly defined by four clusters in terms of sea-level pressure.

The study is in general well written, remaining issues will be corrected in the copy-editing process.

It is of interest to the readership of Atmos. Chem. Phys.

However, I suggest that the authors consider to re-work their study somewhat before it should be accepted for Atmos. Chem. Phys.

(1) The second part of the study, in particular where analysing aerosol effects by weather regime, makes use of the integration without aerosol effects. This is a weak point of the analysis since the reader does not know much about this second integration. Firstly it is necessary to clearly define the differences between the simulations with and without aerosols. Is this really the same model, except that in one the aerosol sources are zero? Or is the model different? Secondly it would be very useful to know whether the two model variants behave comparatively well. The authors could evaluate both model variants in the first part of their study. It would be necessary also that the mean differences in terms of surface radiation and surface temperature are presented. It would be useful to show the geographical patterns of temporal trends of the differences aerosol minus no-aerosol in these two quantities.

(2) The description of the aerosol as a function of weather regime is too long. The authors should consider dropping (or moving to an appendix or supplementary material) many of the plots that are only very superficially discussed and do not help very much the understanding. The conclusions can easily be drawn without this lengthy detail.

Specific comments

p2l25 – MODIS acronym not introduced yet

p3l8 – why "seems to be" only?

p3l14 – Virtually all studies consider of course the interactions (implicitly in interactive simulations and in observational analysis), but do not investigate or analyze these in detail.

p3l16 – probably "analyzing" rather than "establishing" is more what it is

p4l10 – the URL seems to be erroneous

p4l29 – 'subject' rather than 'submitted'

p4l30 – soil only in case of dust presumably

p5l3 – limits in radius or diameter?

p6l32 – MISR data product reference is missing

p7l20 – within → from

p8l20 – is this really a capacity, or wouldn't it rather be very surprising if the regional model deviated a lot from the driving one?p9l1 – correct reference

p9l3 – 0.6 mm day-1 bias translate to a very substantial energy budget problem (of 18 Wm-2 if I'm not mistaken). Is this really acceptable? Where does it come from?

p9l6 – "improved" compared to which reference?

p9l13 – why "also" underestimated? And is it not surprising that a warm bias goes along with a dry bias?

p11l26 – it seems impossible to attribute the biases to specific types

p20l14 – correct section reference

p33/Table 3 – "temperature"; what are the two numbers for ECx?

p34/Table 4 – clarify whether this is for seasonal-mean AOD / NAO index

p37/Fig. 3 – were satellite simulators such as the COSP simulator used for a fair

comparison between the Cloudsat/Calipso and simulated cloud fractions?

p40/Fig. 6 – why is the aerosol concentration not reduced at the domain borders where the boundary condition sets the aerosol to zero?

p43 – is that for seasonal means?

---

## Short Comment (SC1) · 20 Mar 2020

The description of this study is comprehensive, properly structured and well written. I find that it includes a remarkable effort in evaluating the model results with observations, and a very interesting analysis, which improves much our understanding of regional aerosol-climate interactions in the broad Mediterranean region and Europe. I am eager to sea this paper published as part of the ChArMEx special issue. Here is a list of comments and suggestions for minor revisions.

-I wonder whether the aerosol feedbak has any effect on the NAO index. This could be evaluated by comparing the NAO index between the two simulations (AER and NO).

[Figure]

-In a former paper, Nabat et al. (Climate Dynamics, 2015) have demonstrated that atmosphere-ocean coupling enhances aerosol radiative forcing effects, in particular on the surface temperature and sea level pressure. How far could this affect the results on aerosol-induced surface temperature anomalies obtained here with a purely atmospheric model? Is there any possible further aerosol-induced change in the NAO index due to such coupling?

-Moulin et al. (Nature, 1997) correlated winter NAO index and summer dust AOD over the Mediterranean and northeastern tropical Atlantic. Indeed, studies have suggested a delayed effect on dust emissions in semi-arid regions due to the impact of drought on the vegetation, but this is likely not something that the dust emission scheme can take into account.

-There is a contrasted situation in the aerosol load, and especially mineral dust, between summer and spring in the Mediterranean region (e.g. Moulin et al., JGR, 2018; see also Fig. 7) : dust is dominant in summer in the western basin (region D) but much less abundant in the eastern basin (region F), especially during the July and August months with dominant northerly winds (see for instance Fig. 7). On the contrary, intermediate spring and fall seasons are favorable to dust transport in the eastern Mediterranean region, with occurrences of Middle-East dust in fall. As a consequence, I find that there would be some interest in discussing also the spring and fall seasons, at least for dust, and possibly in a Supplement.

-Solar and longwave radiations are significantly variable depending on the region and season: it might be useful to additionally show maps of absolute seasonal values (e.g. in complement of Fig. 4) and give relative differences in % (e.g. in complement to Fig. 5 and in the text).

-Decapitalise northern, northwestern, northeasten, southern, southeastern, eastern, and western.

-Units: I think it is preferred to replace the sign "/" by a space and a negative power as

done in "W m-2" (look for km/h and mm/day).

-Page 5, line 20 : "distribution" may be confusing due to the reference of the vertical dimension in the sentence; I suggest "for the particle size distribution of the emitted dust aerosol (vertical flux)".

-P.5, lines 25-29 : this short paragraph might introduce some doubt on the version used here; I suggest to specify "note for information that [...] coupling of the Mediterranean regional sea, not used in the present study".

-P.6-7, section 2.4: you might specify in the relevant methodological sub-sections the type of aerosol remote sensing product and wavelength(s) considered; AOD at 550 nm from AERONET shown in Figure 7 is probably computed and this is worth a statement in the methodology section; a word on uncertainites of observational products used for model comparison would also be welcome; finally, is there a temporal window selection in model data for comparison to observations? For instance, it is specified in the result section that the comparison with AERONET data is performed on common days, but are the model AOD values a daily or daytime average, or a value at 12UTC?

-P.6-7, section 2.4.1: QuikSCAT is missing in the Methodology sub-section on satellite data.

-P.7, line 2: check citation of Mace and Zhang.

-P.7, section 2.4.2: which version of AERONET products is used? Do you use daily averages and is there a minimum threshold of available measurements in a given day for considering the daily average?

-P.8: I suggest that the sub-section 2.5 Classification in weather regimes should better be shifted after section 2.3 Regional climate simulations since this classification is related to climate model results and not to observations.

-P.8, lines 25-26: reformat the citation "(Christensen and Christensen, 2007)".

-P.9, lines 1-2: check "kotl14" and close the bracket after "Table 3".

-P.9, lines 16-25: it might be useful here specifying the relative radiation biases in % in addition to their absolute values in the different regions (as suggested before).

-P.11, line 17: distributions (plural).

-P.12, line 9: "however" between commas.

-P.15, line 2: "prevents".

-P.15, line 19-20: "a cooling effect" does not seem appropriate with the change by "+0.2°".

-P.21, line 2: "programme" (English spelling, 2 occurrences).

-Table 2: I find that additional columns giving the number of available days, and possibly the overall average AOD for every station would be informative.

-Figure 2: specify in the legend what are the boxes plotted in the upper left map.

-Figure 7: expanding the AOD scale by using a maximum of 0.55 would be hlepful to give a better readability ; not duplicating the ordinate legend in a given line of plots would also allow to expand a bit horizontally the graphs.

-Figure 8: you might note in the legend that the AOD scale is different in each plot.

-Figure 9: I suggest to rotate the figure by 90° counter clockwise in order to expand the graphs.

-Figure 11: for better readability of the plots, I suggest using more contrasted colours and symbols for the filled circles (e.g., black circle and black plus?) and bold characters for legends; you might also vertically expand the graph.

-Figures 12-21: rather use bold characters for all legends; not duplicating axes in a given raw nor a given line in Figs 16-21 would allow expanding the plots.

[Figure]

-Figures 22-23: it might be more intuitive to use red and blue for increased and reduced aerosol impacts, respectively, than the opposite.

[Figure]

---

## Author Comment (AC1) · 19 May 2020

We would like first to thank the reviewers for the evaluation of our work and their positive comments and interesting suggestions. We have addressed all the comments and questions in detail, and clarified the mentioned points. Please find below our point-by-point replies in red. Corrections in the text are indicated in italics (page and line numbers refer to the revised manuscript with highlighted modifications).

**Anonymous Referee #1**

This manuscript investigates the links between aerosols variability modeled by CNRM-ALADIN64 regional climate model with and synoptic atmospheric circulation over the Euro-Mediterranean region, including analysis with respect to the variability of North-Atlantic Oscillation as well as to weather regimes based on persisting meteorological patterns. It is well structured and written and illustrates original and interesting results.I suggest acceptance of the manuscript for publication after taking into consideration the following comments.

R: Thanks for your positive remarks. Note that the model used in this study is finally called CNRM-ALADIN63, as the version is similar to the one published in Euro-CORDEX ALADIN63 simulations. The former name "ALADIN-Climat" is also abandoned in favour of ALADIN for the sake of consistency.

Comments

1) Page 2, lines 15-16: The authors may also consider that Mediterranean cyclones developing in winter and autumn could also affect the dust transport at the Eastern Mediterranean (Flaounas et al., 2015; Georgoulias et al., 2016).

R: Added in the text : "In addition, the formation of Mediterranean cyclones could also affect dust transport over the Eastern Mediterranean in autumn and winter (Flaounas et al., 2015; Georgoulias et al., 2016)." (l20-22 page 2)

2) Page 2,line 18: Please, add a relevant reference

R: Added the reference Israelevitch et al. (2012), line 19 page 2.

3) Page 2, line 19: add a reference. This shift of dust load from Eastern Mediterranean in spring to western Mediterranean in summer has been also shown in a recent study by Marinou et al. (2017) using a satel-lite pure dust product based on CALIPSO.

R: Added the reference to Schepanski et al. (2016) and Marinou et al. (2017), lines 19-20 page 2.

4) Page 10, lines 0-10: I would suggest to discuss shortly the biases in a quantitative manner with respect to MODIS and MISR. Maybe the authors could think of adding a field of the biases

in Figure 6, but this is optional. Of course, in the following section, the authors discuss the AOD biases with respect to station data.

R: We agree the biases should be detailed in this section. Therefore we have added quantitative comparisons of ALADIN and satellite data (MODIS, MISR) in the text (section 3.2.1, see below). As suggested, we have also plotted the map of the AOD bias in ALADIN against MODIS and MISR (Figure S1 below). However, this figure will only be included in Supplementary Material, in order not to overload the paper. The text now reads:

*"However, discrepancies have been found locally, for example in the Benelux and in the Po Valley (see Figure S1), where ALADIN AOD is overestimated compared to MODIS (up to 0.1 in the Po Valley) and especially MISR (up to 0.2). This bias is much smaller than the negative bias in the previous version of the model which did not include nitrate aerosols (Drugé et al., 2019). Annual AOD average over Europe in ALADIN is now similar to MODIS (0.17 for ALADIN and MODIS), but higher than MISR (0.13). Besides, sea-salt aerosols are also probably overestimated over the northern Atlantic Ocean compared to MODIS and MISR, as AOD reaches 0.17 on annual average in this area against only 0.14 for MODIS and 0.12 for MISR. This positive bias is consistent with the surface wind overestimation described in the previous paragraph. Over the Mediterranean where dust particles are prevailing, total AOD simulated by ALADIN (0.18) is in the range of satellite estimates (0.20 for MODIS and 0.16 for MISR). Similar performance is noted over northern Africa (0.27 for ALD-AER, 0.33 for MODIS and 0.34 for MISR)."*

[Figure]

*Figure S2 : Annual average AOD difference (at 550 nm) over the period 2003-2017 between ALADIN and satellite data (MODIS on the left, MISR on the right)*

5) Page 12, lines 2-3: The NAO index used for the analysis would be more consistent if it would be based on ERA-interim which drives ALD-AER rather than the NAO-index provided by NOAA.

Could the authors comment on this issue and justify the use of NOAA instead calculated from ERA-interim?

R: We have indeed used the NAO index provided by NOAA (https://www.cpc.ncep.noaa.gov/products/precip/CWlink/pna/nao.shtml) since this data is ensured to be high-quality data based on observations (Barnston and Livezey, 1987), already used in many published studies, and available without interruption since 1950. Besides, the quality of the ERA-Interim reanalysis has been shown in terms of atmospheric circulation and consistency in time with observations (Dee et al. 2011). In order to check the consistency between ERA-Interim and NOAA data, we have calculated the temporal correlation between this winter NAO index dataset provided by NOAA and winter NAO index calculated with ERA-Interim data. The resulting correlation is 0.90, confirming the consistency between the two datasets. Therefore, we think that using ERA-Interim data would not change our analysis. The justification of the use of NOAA data has been added in the beginning of Section 4.

*"For that purpose, monthly NAO index provided by the National Oceanic and Atmospheric Administration (NOAA, https://www.cpc.ncep.noaa.gov/products/precip/CWlink/pna/nao.shtml) has been used (Barnston and Livezey, 1987) Since the ERA-Interim reanalysis has been shown to be consistent in time with observations and atmospheric circulation (Dee et al., 2011), this NAO index data must be consistent with the ERA-Interim reanalysis and therefore the ALADIN simulations."*

6) Page 12, lines 16-18: The justification provided by the authors is very reasonable. I think that a strong support on all this discussion would be provided by plotting near surface wind vectors along with model AOD separately for the positive phase and the negative NAO phase. This is, however, a suggestion which could be only optionally considered by the authors.

R: We agree with the reviewer's suggestion, and we have plotted the average surface wind vectors along with AOD and sea level pressure in this new figure, separately for the positive and negative phases of NAO. This figure has been included in the manuscript (Figure 10), and the following text has been added in the new version (section 4.1 pages 13-14).

*"Figure 10 shows both the AOD anomalies and the average circulation in the surface (wind and sea level pressure) respectively for the positive and negative phases of NAO. In the positive phase, both the low pressures over Iceland (beyond the northern limit of the domain) and the high pressures in the Azores are reinforced, the latter also reinforcing northeastern winds over northwestern Sahara following the geostrophic wind circulation. In the negative phase, both action centres move south, thus increasing wind speed over the Atlantic Ocean between 30 and 40° N, but weakening winds over the Sahara."*

[Figure]

*Figure 10 : Averaged atmospheric circulation (sea level pressure in hPa, purple lines and surface wind, wind barbs in black) with AOD anomalies (colors) in winter (DJF, top) and in summer (bottom), respectively for the positive (left) and negative (right) phase of NAO*

7) Page 13, lines 19-20: Here it is mentioned that less rainfall is noted in summer under the positive phase of NAO but according to Blade et al. (2011) during high NAO summers, when strong anticyclonic conditions and suppressed precipitation prevail over the UK, the Mediterranean region instead is anomalously wet (see for example their Figure 10). Could you check in your analysis the regions that less rainfall is noted in summer under the positive phase of NAO?

[Figure]

*Figure S3 : Averaged precipitation anomalies (mm/day) during the positive (left) and negative (right) phase of NAO simulated by ALADIN in summer (JJA).*

R: We have plotted the anomalies of precipitation under the positive and negative phases of NAO (Fig. S3). This figure shows that the decrease of rainfall in the positive phase concerns only western and northern Europe, and not really southern Europe and the Mediterranean. The positive correlation between sulfate AOD and NAO index in this area is consequently not due to the decrease in precipitation. The text is corrected as follows:

*"In the positive phase of NAO in summer, the Mediterranean region is wetter than average (Bladé et al. 2012), with a slight positive anomaly in precipitation (Fig. S3). Thus the increase in sulphate AOD could be due to an increase in relative humidity in the lower troposphere, which enhances aerosol extinction of hydrophilic aerosol species such as sulphate."*

8) Page 20, lines 15-18: Could you make a course estimate of the impact of using monthly instead of daily AOD climatology in RCM simulations? I guess this can have a stronger impact on SSR but the impact on near surface temperature is maybe trivial.

R: This is a very interesting question, but which will deserve a full article to be answered. Aerosol effects may indeed be non linear because essentially of the interactions between clouds and aerosols that are shown in the present study. First elements of response can be found in Nabat et al. (2015b), who have shown an impact of the use of interactive aerosols instead of AOD climatology on surface radiation (between 2 and 5 W/m² on average) and surface temperature (between 0.2 and 0.4 °C on average) during summer 2012. However, we need longer simulations (at least several years) to have a precise answer, and we hope to do it in a future study. The following sentence has been added:

*"Nabat et al. (2015b) have shown that during summer 2012 the use of interactive aerosols instead of AOD climatologies could lead to differences in surface radiation of about 5 W/m² and in surface temperature of about 0.4°C over the Mediterranean region."*

9) Figure 1: Please describe in figure caption the numbering.

R: Added. The caption is now as follows:

*"AERONET and BSRN stations used in this study have been added with coloured crosses and circles respectively (See Table 2 for the names of AERONET and BSRN stations), as well as the nine subregions in which they are gathered (A:1-9, B:10-18, C:19-27, D:28-36, E:37-45, F:46-54, G:55-63, H:64-72 and I:73-81)."*

10) Figure 2: QuikSCAT is presented here but not described. Please add in Section 2 a description of QuikSCAT dataset used here for model evaluation of sea winds. Discuss also how the spatial resolution of the model and observation data compares. You may also think also of presenting a comparison of ALADIN surface wind with ERA-Interim for consistency with the SLP comparison.

R: A description of QuikSCAT data has been added in Section 2.5.1 Satellite data. In order to evaluate surface wind in ALADIN simulations, we have chosen to use QuikSCAT data rather than ERA-Interim because resolution does matter for this parameter over the Mediterranean (Herrmann et al. 2011) contrary to sea level pressure which is smoother in space. Several studies (Ruti et 2007, Chronis et al. 2010, Herrmann et al. 2011) have shown the performance of QuikSCAT data over the Mediterranean region.

*"As far as surface wind is concerned, QuikSCAT data provide satellite observations over the sea at 0.25° resolution. The ability of this instrument to retrieve the in-situ variability of both wind direction and speed has been shown by Ruti et al. (2007). The high resolution makes it suitable for studies over the Mediterranean (Chronis et al. 2010, Herrmann et al. 2011). The version used here is the level 3 dataset, similar as the one used in the previous evaluation of ALADIN carried out in Nabat et al. (2015b)."*

11) Figure 3: The low cloud and total cloud differences plot is of pure quality and not really informative because of the gaps. Discuss also how the spatial resolution of the model and observation data compares.

R: We agree the figures showing low and total cloud differences were not very informative, mainly because of the gaps due to the fact that this product has been built directly on the ALADIN grid at 50km depending on the tracks of the CloudSAT and CALIPSO instruments. In order to have a gridded product as those used for the other variables, we have interpolated this product on all the grid points using the ncl *poisson_grid_fill* function (https://www.ncl.ucar.edu/Document/Functions/Built-in/poisson_grid_fill.shtml). Besides, the sensitivity of the lidar in this lidar-radar combined product to obtain cloud fraction has been adjusted to the release of the CloudSAT GEOPROF products. This change has consequences on the comparison between ALADIN and CloudSAT-CALIPSO shown in Figure 11. Averages in Table 3 have also been modified. The underestimation of cloud cover in winter over the Mediterranean is more pronounced. The text in sections 2.4.1 and 3.1 has been modified.

*Section 2.5.1: For model comparison purpose, a cloud fraction is computed from this observational data set in each ALADIN model grid point as the fraction of the grid covered by a*

*cloud detected in radar geometrical profile where the cloud mask is higher than 20 (corresponding to less than 16 % of false detection) or when the lidar cloud fraction exceed 10 % in a given bin. These thresholds differ from the values initially proposed and validated by Mace et al. (2009) because of the use of the release 05 of the CloudSat GEOPROF products with specific tests performed on our domain.*

*Section 3.1: "Besides, cloud cover is significantly improved in Europe compared to the previous version of the model, as the bias is only -4% on average."*

[Figure]

**c) Total cloud cover**

**d) Low cloud cover**

*Extract of Figure 3 (c and d) : Winter (DJF, left) and summer (JJA, right) average differences between ALADIN and observations (...) for cloud cover (%, 2006-2011, total fraction in c, low fraction in d).*

Minor Corrections

1) Page 2, line 9: Please, specify which season.

R: "In this season" has been replaced by "in spring and summer".

2) Page 12, lines17: easterly winds instead of eastern winds.

R: Corrected.

**Anonymous Referee #2**

Nabat et al. present firstly a new model version of a regional climate model that has a number of revised parameterisations compared to a previous version. An evaluation with multiple observational datasets is presented. This evaluation is a bit hampered by the fact that no direct comparison is presented between the new model version and an older one, even if often the text compares the skill of the previous version (unknown to the reader unless they carefully studied the former papers by the authors). In particular the aerosol distributions and temporal variability are compared to satellite retrievals and surface remote sensing. The models shows a rather remarkable skill both for the geographical distribution and the annual cycles of aerosol optical depths. The bulk of the manuscript is a lengthy analysis on how aerosols are simulated differently for different weather conditions as firstly defined by the NAO index (presumably as seasonal averages, the text needs to be clarified on the temporal resolution of the analysis), and secondly defined by four clusters in terms of sea-level pressure.

The study is in general well written, remaining issues will be corrected in the copy-editing process.

It is of interest to the readership of Atmos. Chem. Phys. However, I suggest that the authors consider to re-work their study somewhat before it should be accepted for Atmos. Chem. Phys.

R: Thanks for the evaluation of our work and the positive comments. With regards to the comparison between the new model version and an older one, we have clarified in the text that the new model version is the current version used in this study named CNRM-ALADIN63, while the older one refers to the version 5 of ALADIN used in CNRM-RCSM4 (Sevault et al. ,2014; Nabat et al., 2015a) and in CNRM-RCSM5 (Nabat et al. 2015b). For ease of comparison, we have also added in Tables 3 and 4 the values calculated with a similar simulation carried out with the same ALADIN version 5 used in Nabat et al. (2015b), in order to justify the improvement brought by the new version CNRM-ALADIN63. This is noted in the beginning of Section 3.1.

*"This evaluation of the new version 6.3 of the ALADIN model is also to be compared with a similar work carried out with the previous version 5 of ALADIN (Sevault et al. 2014, Nabat et al., 2015a, b). In Tables 3 and 4, biases calculated with an ALADIN simulation (1979-2012) carried out with the version 5.3 used in Nabat et al. (2015a) have been added for ease of comparison with the new version 6.3."*

(1) The second part of the study, in particular where analysing aerosol effects by weather regime, makes use of the integration without aerosol effects. This is a weak point of the analysis since the reader does not know much about this second integration. Firstly it is necessary to clearly define the differences between the simulations with and without aerosols. Is this really the same model, except that in one the aerosol sources are zero? Or is the model different? Secondly it would be very useful to know whether the two model variants behave comparatively well. The authors could evaluate both model variants in the first part of their study. It would be necessary also that the mean differences in terms of surface radiation and surface temperature

are presented. It would be useful to show the geographical patterns of temporal trends of the differences aerosol minus no-aerosol in these two quantities.

R: Both simulations ALD-AER and ALD-NO have been carried out with the same model (CNRM-ALADIN63). The only difference is indeed the absence of all aerosols in ALD-NO, in other words the aerosol optical depth is set to zero in ALD-NO (this has been clarified in the revised version of the manuscript). We agree this difference could have an impact on mean climate, notably on radiation and surface temperature. However, an evaluation of the ALD-NO simulation would be like assessing the effects of aerosols on mean climate, which is not the scope of the present paper as it has already been elaborated in two previous studies (Nabat et al. 2015a, 2015b). These two previous studies had indeed the same methodology of comparing simulations with and without aerosols. Besides, we believe that the main biases found in ALD-AER (the overestimation of precipitation in winter in Europe by 34%, the warm bias in summer in Europe of +1.3°C associated with an underestimation of precipitation by 32%) are an order of magnitude higher than the mean aerosol effects on regional climate in Europe.

*"Note that the ALD-NO simulation, which is similar to ALD-AER apart from the aerosols (AOD is set to zero in ALD-NO), is not evaluated here, since such a couple of simulations had already been the focus of two previous studies (Nabat et al., 2015a, b)*

(2) The description of the aerosol as a function of weather regime is too long. The authors should consider dropping (or moving to an appendix or supplementary material) many of the plots that are only very superficially discussed and do not help very much the understanding. The conclusions can easily be drawn without this lengthy detail.

R: We agree the figures presenting the aerosol effects as a function of weather regime were probably too numerous, which made the understanding of this section difficult. Therefore, following the reviewer's suggestion, we have decided to replace the three different figures presenting probability distribution functions for five subregions in winter, by one single figure presenting the same probability distribution functions for only three regions (EURNW, ALPS and EURSW, Figure 17). The two other regions (EURN and EURSE) have been moved to Supplementary Material. The same presentation has been adopted for summer (Figure 18), which removes a total of four figures from the paper. The text has been adapted to this new presentation, focusing mainly on the three regions kept in the main text.

Specific comments

p2l25 – MODIS acronym not introduced yet
R: The acronym has been defined here: "the MODerate resolution Imaging Spectroradiometer".

p3l8 – why "seems to be" only?
R: "Seems to be" is replaced by "is" since several references (Gkikas et al. 2013; Nabat et al., 2015a; Schepanski et al. 2016) justify this affirmation.

p3l14 – Virtually all studies consider of course the interactions (implicitly in interactive simulations and in observational analysis), but do not investigate or analyze these in detail.
R: The idea here was to point out the fact that climate-aerosol interactions are not treated at high temporal frequency in regional climate simulations. The text has been clarified.
*"Most of climate studies based on regional climate simulations already published only consider these interactions at yearly or seasonal time scales, while the daily time scale would be needed to better understand these interactions."*

p3l16 – probably "analyzing" rather than "establishing" is more what it is
R: Corrected.

p4l10 – the URL seems to be erroneous
R: Corrected: http://www.umr-cnrm.fr/spip.php?article1092&lang=en

p4l29 – 'subject' rather than 'submitted'
R: Corrected.

p4l30 – soil only in case of dust presumably
R: Soil characteristics are only for dust emissions. Corrected in the text.

p5l3 – limits in radius or diameter?
R: Diameter, added in the text.

p6l32 – MISR data product reference is missing

p7l20 – within→from
R: Corrected.

p8l20 – is this really a capacity, or wouldn't it rather be very surprising if the regional model deviated a lot from the driving one?
R: Indeed it would have been very surprising that ALD-AER deviates a lot from ERA-Interim, but it is worth checking that the driving by lateral boundaries and spectral nudging is correctly applied. Note that the way RCMs reproduce the large-scale pattern of their driving model can vary from one model to another and depends strongly on the temporal scale  (Sanchez-Gomez et al. 2009, Sanchez-Gomez and Somot 2018).

p9l1 – correct reference
R: Corrected (Kotlarski et al. 2014).

p9l3 – 0.6 mm day-1 bias translate to a very substantial energy budget problem (of 18Wm-2 if I'm not mistaken). Is this really acceptable? Where does it come from?

R: We agree that we have a significant bias in precipitation in winter in Europe, namely 0.6 mm day$^{-1}$, which represents 34% of the precipitation. However, this bias has no impact on radiative budget, since the shortwave radiation at the top of the atmosphere is very close to satellite data (+0.1 W m$^{-2}$). It would deserve further analyses to understand the origin of this underestimation of precipitation, which is out of the scope of the present study.

p9l6 – "improved" compared to which reference?

R: Compared to the previous version of the model (added in the text).

p9l13 – why "also" underestimated? And is it not surprising that a warm bias goes along with a dry bias?

R: We agree, the text has been modified:

"It is combined with an underestimation of summer precipitation in Europe"

p11l26 – it seems impossible to attribute the biases to specific types

R: We agree that the evaluation of aerosol optical depth presented in this section only refers to the total aerosol load, and not specifically to each aerosol type. However, given that some regions are characterized with specific aerosol types (for example desert dust in the Sahara), we can make assumptions about the origin of these biases. We have thus modified the text to modify our conclusions in that sense. The text is now (lines 8-11 page 13):

*"Some discrepancies have also been emphasized, notably in spring in Northern Europe likely due to an overestimation of nitrates and in summer in the Atlantic and in Southeastern Europe presumably because of an underestimation of the dust transport."*

p20l14 – correct section reference

R: Corrected (it was line 24 and not 14).

p33/Table 3 – "temperature"; what are the two numbers for ECx?

R: These two numbers are the minimum and maximum bias among Euro-CORDEX models (added in the caption of Table 3).

p34/Table 4 – clarify whether this is for seasonal-mean AOD / NAO index

R: All values in Table 5 (AOD and NAO index) have been calculated with seasonal means (DJF on the left, JJA on the right). This has been clarified in the caption of Table 5.

p37/Fig. 3 – were satellite simulators such as the COSP simulator used for a fair comparison between the Cloudsat/Calipso and simulated cloud fractions?

R: Unfortunately the COSP simulator was not available in these simulations, but could be included in future simulations. However, as described in Section 2.5.1, the product

Cloudsat/CALIPSO used here has been directly built on the ALADIN grid, in order to take into account the exact location of the radar and the lidar.

p40/Fig. 6 – why is the aerosol concentration not reduced at the domain borders where the boundary condition sets the aerosol to zero?
R: The ALADIN domain includes a relaxation zone of 8 points around the domain which is not shown in the figures. That is the reason why the decrease in aerosol concentration near the domain borders is not really visible (except a little in the South as you can see when you compare ALADIN and MODIS).

p43 – is that for seasonal means?
R: Yes, added in the caption.

**François Dulac**

[francois.dulac@cea.fr](mailto:francois.dulac@cea.fr)

The description of this study is comprehensive, properly structured and well written. I find that it includes a remarkable effort in evaluating the model results with observations, and a very interesting analysis, which improves much our understanding of regional aerosol-climate interactions in the broad Mediterranean region and Europe. Iam eager to sea this paper published as part of the ChArMEx special issue. Here is a list of comments and suggestions for minor revisions.

-I wonder whether the aerosol feedback has any effect on the NAO index. This could be evaluated by comparing the NAO index between the two simulations (AER and NO).
R: This is an interesting question, however it cannot be answered with these simulations for two reasons. First we have used spectral nudging in our simulations, which drives the large scale circulation and probably the NAO index. Secondly, our domain is too small to calculate the NAO index directly in ALADIN (notably it does not include Iceland).

-In a former paper, Nabat et al. (Climate Dynamics, 2015) have demonstrated that atmosphere-ocean coupling enhances aerosol radiative forcing effects, in particular on the surface temperature and sea level pressure. How far could this affect the results on aerosol-induced surface temperature anomalies obtained here with a purely atmospheric model? Is there any possible further aerosol-induced change in the NAO index due to such coupling?
R: We agree that ocean-atmosphere coupling is important to take into account to estimate aerosol radiative forcing over the Mediterranean, but due to computational cost it was not possible to include this coupling in the simulations used in our study. However as shown in Nabat et al. (2015a) the main effects of ocean-atmosphere coupling on the climate-aerosol interactions concern surface temperature over ocean and over coastal regions, as well as hydrological cycle because of the modification of evaporation. In our study, we focus more on radiative impacts and changes on land surface temperature, potential impacts on precipitation are not discussed because of this absence of ocean-atmosphere coupling.

-Moulin et al. (Nature, 1997) correlated winter NAO index and summer dust AOD over the Mediterranean and northeastern tropical Atlantic. Indeed, studies have suggested a delayed effect on dust emissions in semi-arid regions due to the impact of drought on the vegetation, but this is likely not something that the dust emission scheme can take into account.
R: Indeed this effect of winter NAO index on summer dust AOD through vegetation is an interesting hypothesis, but it cannot be simulated by ALADIN since there is no interactive vegetation yet (added in the description of the model).
*"Nevertheless the model configuration does not include interactive vegetation which could impact dust emissions (Pierre et al. 2012)."*

-There is a contrasted situation in the aerosol load, and especially mineral dust, be-tween summer and spring in the Mediterranean region (e.g. Moulin et al., JGR, 2018; see also Fig. 7) : dust is dominant in summer in the western basin (region D) but much less abundant in the eastern basin (region F), especially during the July and August months with dominant northerly winds (see for instance Fig. 7). On the contrary, intermediate spring and fall seasons are favorable to dust transport in the eastern Mediterranean region, with occurrences of Middle-East dust in fall. As a consequence,I find that there would be some interest in discussing also the spring and fall seasons,at least for dust, and possibly in a Supplement.

R: We agree that spring and fall may be of interest to study the modulation of aerosol radiative effects by atmospheric circulation. However, this would require further analyses that would increase the length of the paper, which had already been judged too long by the second reviewer.

-Solar and longwave radiations are significantly variable depending on the region and season: it might be useful to additionally show maps of absolute seasonal values (e.g.in complement of Fig. 4) and give relative differences in % (e.g. in complement to Fig.5 and in the text).

R: We agree it would be interesting to have these figures in % in addition to the absolute values, but this would lengthen the paper which already has many figures. However, to follow this suggestion, we have added this figure in Supplementary Material (Figure S1), and an estimation of the percentage in the text (Section 3.1). Note that we have also added in this figure an evaluation of surface radiation against SARAH data set for shortwave (a description of SARAH data has been added in Section 2.5.1) and CERES data set for longwave.

*"Both for SW and LW radiation at the TOA, the weak remaining bias over Europe and the Mediterranean represents less than 5% of the total upward radiation (Figure S1)."*

[Figure]

*Figure S1 : Winter (DJF, left) and summer (JJA, right) average relative differences (in \%) between ALADIN and satellite data for shortwave (SW) and longwave (LW) radiation at the top of the atmosphere (upward fluxes, a for SW, b for LW) and at the surface (downward fluxes, c for SW, d for LW). Satellite data used here are CERES (2000-2016) for a, b and d, as well as SARAH (1983-2015) for c.*

-Decapitalise northern, northwestern, northeasten, southern, southeastern, eastern, and western.

R: Corrected.

-Units: I think it is preferred to replace the sign "/" by a space and a negative power as done in "W m-2" (look for km/h and mm/day).

R: Corrected.

-Page 5, line 20 : "distribution" may be confusing due to the reference of the vertical dimension in the sentence; I suggest "for the particle size distribution of the emitted dust aerosol (vertical flux)".

R: Corrected.

-P.5, lines 25-29 : this short paragraph might introduce some doubt on the version used here; I suggest to specify "note for information that [...] coupling of the Mediterraneanregional sea, not used in the present study".

R: We agree, this information has been clarified.

-P.6-7, section 2.4: you might specify in the relevant methodological sub-sections the type of aerosol remote sensing product and wavelength(s) considered; AOD at 550 nm from AERONET shown in Figure 7 is probably computed and this is worth a statement in the methodology section; a word on uncertainites of observational products used for model comparison would also be welcome; finally, is there a temporal window selection in model data for comparison to observations? For instance, it is specified in the results section that the comparison with AERONET data is performed on common days, butare the model AOD values a daily or daytime average, or a value at 12UTC?

R: AERONET AOD at 550 nm is computed from wavelengths at 440 or 500 nm (depending on availability), and Angstrom exponent. Daily AOD at 550 nm can then be compared to daily AOD averages of ALADIN simulations at the same wavelength (added in Section 2.4.2). As mentioned already at the beginning of Section 3.2.2., we ensure to keep the same days in AERONET and in ALADIN for each station, which represents a substantial effort to have a fair comparison given that many climate simulations are only evaluated at the monthly scale. However, there is no temporal window selection smaller than the day in model data. The uncertainty of AOD given by AERONET stations is ±0.01 (Eck et al. 1999).

*"For each of them, AERONET AOD at 550 nm is computed from wavelengths at 440 or 500 nm (depending on availability), and Angstrom exponent at daily frequency. Daily AOD at 550 nm can then be compared to daily AOD averages of ALADIN simulations at the same wavelength."*

-P.6-7, section 2.4.1: QuikSCAT is missing in the Methodology sub-section on satellite data.

R: The description of QuikSCAT has been added in Section 2.5.1.

*"As far as surface wind is concerned, QuikSCAT data provide satellite observations over the sea at 0.25° resolution. The ability of this instrument to retrieve the in-situ variability of both wind direction and speed has been shown by Ruti et al. (2007). The high resolution makes it suitable for studies over the Mediterranean (Chronis et al. 2010, Herrmann et al. 2011). The version used here is the level 3 dataset, similar as the one used in the previous evaluation of ALADIN carried out in Nabat et al. (2015b)."*

-P.7, line 2: check citation of Mace and Zhang.
R: Corrected: Mace and Zhang (2014).

-P.7, section 2.4.2: which version of AERONET products is used? Do you use daily averages and is there a minimum threshold of available measurements in a given day for considering the daily average?
R: We have used AERONET version 3 daily averages, which are directly provided by AERONET (added in the text). We trust AERONET daily products to be representative enough of the daily means, as we do not have access to the number of values considered in the calculation of the daily average.

-P.8: I suggest that the subsection 2.5 Classification in weather regimes should better be shifted after section 2.3 Regional climate simulations since this classification is related to climate model results and not to observations.
R: We agree, the subsection "Classification in weather regimes" has been shifted as Section 2.4 before Observations.

-P.8, lines 25-26: reformat the citation "(Christensen and Christensen, 2007)".
R: Corrected.

-P.9, lines 1-2: check "kotl14" and close the bracket after "Table 3".
R: Corrected.

-P.9, lines 16-25: it might be useful here specifying the relative radiation biases in % in addition to their absolute values in the different regions (as suggested before).

-P.11, line 17: distributions (plural).
R: Corrected.

-P.12, line 9: "however" between commas.
R: Corrected.

-P.15, line 2: "prevents".
R: Corrected with an "s" in "high pressures": "high pressures prevent".

-P.15, line 19-20: "a cooling effect" does not seem appropriate with the change by"+0.2∘".
R: Corrected, it was -0.2°C.

-P.21, line 2: "programme" (English spelling, 2 occurrences).
R: Corrected.

-Table 2: I find that additional columns giving the number of available days, and possibly the overall average AOD for every station would be informative.
R: The information

-Figure 2: specify in the legend what are the boxes plotted in the upper left map.
R: Added.

-Figure 7: expanding the AOD scale by using a maximum of 0.55 would be helpful to give a better readability ; not duplicating the ordinate legend in a given line of plots would also allow to expand a bit horizontally the graphs.
R: As suggested, in order to enlarge the graphs in Figure 7, we have used a maximum of 0.55 instead of 0.7, and removed the title of the y-axis ("AOD") when possible.

-Figure 8: you might note in the legend that the AOD scale is different in each plot.
R: Added: "Note that the AOD scale is adapted to each region."

-Figure 9: I suggest to rotate the figure by 90∘counter clockwise in order to expand the graphs.
R: We have swapped rows and columns to enlarge the graphs.

-Figure 11: for better readability of the plots, I suggest using more contrasted colours and symbols for the filled circles (e.g., black circle and black plus?) and bold characters for legends; you might also vertically expand the graph.
R: As suggested by the reviewer we have replaced green and purple circles by filled black circles and black crosses respectively. We have also enlarged the size of the labels.

-Figures 12-21: rather use bold characters for all legends; not duplicating axes in a given raw nor a given line in Figs 16-21 would allow expanding the plots.
R: We have replaced the six figures 16 to 21 by two figures with enlarged graphs, we have also avoided the duplication of axes for the same rows and lines when possible.

-Figures 22-23: it might be more intuitive to use red and blue for increased and reduced aerosol impacts, respectively, than the opposite.
R: We have chosen these colors because blue is thus associated to a cooling and/or a decrease in surface radiation, and/or red to a warming or an increase in surface radiation. In order to make

the figure more understandable, we have replaced the cloud cover average by the AOD anomaly in the background.